

# A novel metabolism-related gene signature in patients with hepatocellular carcinoma

Bin Ru[1,*], Jiaqi Hu[1,2,3,*], Nannan Zhang[1] and Quan Wan[1]

[1] Department of Pain Management, Zhejiang Provincial People's Hospital (Affiliated People's Hospital, Hangzhou Medical College), Hangzhou, Zhejiang, China
[2] State Key Laboratory of Medical Neurobiology and MOE Frontiers Center for Brain Science, Fudan University, Shanghai, China
[3] Department of Physiology and Neurobiology, School of Life Sciences, Fudan University, Shanghai, China
[*] These authors contributed equally to this work.

## ABSTRACT

Hepatocellular carcinoma (HCC) remains a global challenge as it is the sixth most common neoplasm worldwide and the third leading cause of cancer-related death. A key feature of HCC is abnormal metabolism, which promotes cancer cell proliferation, survival, invasion, and metastasis. However, the significance of metabolism-related genes (MRGs) in HCC remains to be elucidated. Here, we aim to establish a novel metabolism-related prognostic signature for the prediction of patient outcomes and to investigate the value of MRG expression in the prognostic prediction of HCC. In our research, a Metabolism-Related Risk Score (MRRS) model was constructed using 14 MRGs (DLAT, SEPHS1, ACADS, UCK2, GOT2, ADH4, LDHA, ME1, TXNRD1, B4GALT2, AK2, PTDSS2, CSAD, and AMD1). The Kaplan-Meier curve confirmed that the MRRS has a high accuracy in predicting the prognosis of HCC patients ($p < 0.001$). According to the MRRS model, the area under the curve (AUC) values for predicting the prognosis of patients with hepatocellular carcinoma at 1, 3, and 5 years reached 0.829, 0.760, and 0.739, respectively. Functional analyses revealed that signaling pathways associated with the cell cycle were largely enriched by differential genes between high and low-risk groups. In addition, dendritic cells (DCs) ($p < 0.001$), CD4+ T cells ($p < 0.01$), CD8+ T cells ($p < 0.001$), B cells ($p < 0.001$), neutrophils ($p < 0.001$), macrophages ($p < 0.001$) had a higher proportion of infiltrates in high-risk populations. Low GOT2 expression is associated with poor prognosis in patients with hepatocellular carcinoma. Knockdown of GOT2 significantly increased the migration capacity of the Huh7 and MHCC97H hepatocellular carcinoma lines. Our research reveals that GOT2 is negatively related to the survival of patients with hepatocellular carcinoma and GOT2 may contribute to tumor progression by inhibiting the ability of tumor cells to migrate.

Corresponding author
Quan Wan,
wanquan1989114@126.com

## INTRODUCTION

The incidence of liver cancer is increasing worldwide (*Llovet et al., 2021*; *Villanueva, 2019*). Liver cancer is the sixth most common malignancy and the third leading cause of cancer-related death globally. According to the latest statistics, there are 41,210 new cases of liver cancer and 29,380 deaths due to liver cancer in the United States in 2022 (*Siegel et al., 2023*). Hepatocellular carcinoma (HCC) patients account for more than 90% of liver cancer cases, and with the implementation of targeted and immunotherapy, the life expectancy of patients with hepatocellular carcinoma has increased (*Llovet et al., 2018*; *Zucman-Rossi et al., 2015*; *Schulze et al., 2015*). These treatments have a good therapeutic effect in patients with early-stage hepatocellular carcinoma. Due to chemotherapy tolerance and insensitivity to radiotherapy, the choice of effective treatments for patients with advanced hepatocellular carcinoma is seriously affected, resulting in poor prognosis (*Llovet, Burroughs & Bruix, 2003*). In the United States, the 5-year survival of patients with HCC is only 18% (*Jemal et al., 2017*). Traditional prognosis prediction systems for cancer patients, such as TNM staging, are increasingly difficult to cover the diversity of clinical features of HCC patients (*Shao et al., 2016*). The development of new prognostic models at multiple levels will help to better distinguish between different types of HCC patients (*Shibata, 2021*). Recent studies have demonstrated favorable outcomes in discriminating between HCC and predicting HCC prognosis, through the creation of multiple test-based indicators (*Luo et al., 2022*; *Xie et al., 2022*). This facilitates better extraction of features from different types of patients to facilitate more effective treatment of different types of patients.

Abnormal metabolism is one of the characteristics of HCC (*Hanahan & Weinberg, 2011*). During the malignant transformation of cells, metabolism usually undergoes drastic changes. Tumor cells regulate metabolism, promoting energy production and accumulation (*Cheng et al., 2018*). Hepatocellular carcinoma cells have elevated levels of metabolism to maintain their high metabolic levels for plasma membrane synthesis and energy production (*Sangineto et al., 2020*). Oncogenic signaling pathways, including B-Raf kinase (BRAF) and epidermal growth factor receptor (EGFR), drive dysregulation of fatty acid metabolism, affecting membrane composition and saturation to regulate tolerance to reactive oxygen species (ROS) and cancer cell survival (*Talebi et al., 2018*; *Gimple et al., 2019*; *Bi et al., 2019*). In addition, recent studies have shown that alterations in metabolism in tumor cells are able to promote tumor invasion and metastasis, regulate oxidative stress, and provide energy under various cellular stress situations (*Pascual et al., 2017*; *Rohrig & Schulze, 2016*; *Ladanyi et al., 2018*). Given that metabolism abnormalities play an important role in tumor progression, it has not been explored whether metabolism genes can be analyzed to build a prognostic model for HCC patients.

The role of metabolism in the regulation of immune cells has recently attracted widespread attention. Research evidence in several types of solid tumors shows the importance of metabolic reprogramming of immune cells in tumors, suggesting a new strategy for the treatment of HCC patients (*Zhang et al., 2018*). The function of immune cells in the HCC tumor microenvironment is closely related to abnormal metabolism

(*Hu et al., 2020*). However, whether the expression level of MRGs in HCC patients affects immune cell infiltration in the tumor microenvironment remains to be explored.

In our research, we developed a novel prognostic prediction model based on MRGs, which is named "Metabolism-Related Risk Score" (MRRS), to predict the prognosis and survival of HCC patients. The immune-related components of different risk groups were analyzed, and the results showed that there were significant differences in the tumor immune microenvironment between the two groups. The evaluation of our predictive typing model shows great potential to guide the classification and treatment of HCC patients, demonstrating its clinical value and significance. Currently, a variety of genes and proteins that regulate metabolism have been identified, including GOT2, which has been reported to be involved in metabolism (*Liu et al., 2022*). However, the effect of this metabolism-related gene on the development and progression of hepatocellular carcinoma is unclear. Our study preliminarily explored the relationship between GOT2 and the prognosis of patients with hepatocellular carcinoma and preliminarily explored its effect on the migration ability of hepatocellular carcinoma.

## METHODS

### Collection and processing of gene expression data and clinical information in HCC patients

Gene expression data and corresponding clinical information of 374 tumor tissues and 50 normal tissues of 374 patients with hepatocellular carcinoma as of March 03, 2022, were obtained from TCGA (https://portal.gdc.cancer.gov/repository). Of these, 44 patients were removed due to incomplete clinical data. Therefore, tissue gene expression data and complete follow-up information from the remaining patients ($n = 330$) were incorporated into our training dataset for further analysis. The test dataset used for validation was gene expression data and clinical information from an additional 231 tumor samples obtained from the ICGC portal (https://dcc.icgc.org/projects/LIRI-JP). These samples were mainly from Japan (*Fujimoto et al., 2016*). Since this data is all available online and comes with permission to use it, no additional ethical approval is required. The current study follows the TCGA and ICGC data access policies and publication guidelines. The flowchart of this research is shown in Fig. 1.

### Construction and validation of metabolism-related risk scores

Utilize the "limma" R package to determine the DEGs associated with metabolism between tumor and normal tissues, the false discovery rate (FDR) in the TCGA cohort < 0.05. Univariate Cox analysis of overall survival (OS) to screen for prognostically relevant MRGs, and visualize it with a forest plot. The intersection of metabolism-related DEGs with prognostic genes is demonstrated by a Venn diagram and visualized by a heatmap. An interactive network for generating prognostic DEGs from the Search Tool for the Retrieval of Interacting Genes (STRING) database (https://string-db.org). We use the LASSO Cox regression analysis to build a prognostic model that minimizes the risk of overfitting by performing the "glmnet" function of the R package. The penalty parameter (λ) of the model is determined by tenfold cross-validation following the minimum criterion (*i.e.*, the

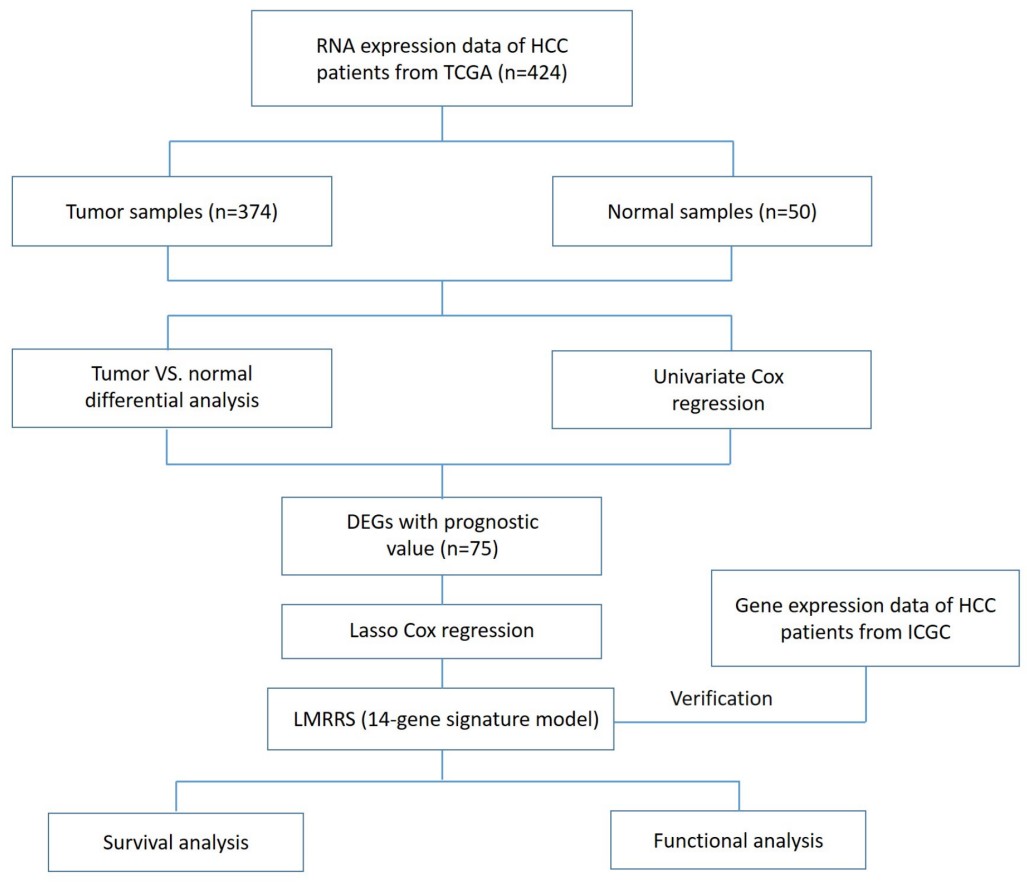

**Figure 1 Flow chart of data collection and analysis in the present study.**

$\lambda$ value corresponding to the lowest partial likelihood bias). Subsequently, the patient's risk score is calculated based on gene expression and the corresponding Cox regression coefficient as follows: score = sum (expression of each gene × corresponding coefficient) (Table S1). Patients were then divided into high-risk and low-risk groups based on median risk score values. Based on the expression of gene signatures, PCA is performed using the "prcomp" function of the "stats" R package. T-SNE and PCA analysis using the prcomp function in the "Rtsne" package and the "stats" package to explore the distribution of high- and low-risk groups. Kaplan–Meier survival curve and time-dependent ROC curve analysis were applied to compare survival between the above two groups to assess the predictive accuracy of gene signatures.

## Functional enrichment analysis of HCC patients classified based on MRRS

Classification of HCC patients based on MRRS. Patients between high- and low-risk groups were analyzed for gene ontology (GO) and Kyoto Encyclopedia of Genes and Genomes (KEGG) pathways using the "clusterProfiler" R package in the high- and low-risk groups

(29,31). The GO term and KEGG pathway with a *p*-value of <0.05 were statistically significant.

## Cell culture

The human hepatocellular carcinoma cell lines Huh7 and MHCC97H were purchased from the American Type Culture Collection (Rockville, MD, USA). Cell lines were all maintained at 5% $CO_2$ at 37 °C and cultured in DMEM medium (Gibco, Waltham, MA, USA) supplemented with 10% and 13% FBS (Gibco, Waltham, MA, USA).

## Small interfering (siRNA) transfection

Tsingke biological technology offers GOT2 small interference RNA (siGOT2-1#, siGOT2-2#, siGOT2-3#) and non-target small interference RNA (siNC). The siGOT2 sequence was designed by Tsingke Biologics. Follow the manufacturer's instructions for transfection in Opti-MEM medium (Gibco, Waltham, MA, USA) using RNAiMax Transfection Reagent (Invitrogen, Carlsbad, CA, USA). After stable transcription, collect cells for the next step of the experiment.

## RNA extraction and qRT-PCR

Total RNA was isolated and extracted from HEK293 cells using TRIzol reagent (Invitrogen, Carlsbad, CA, USA) and detected using a NanoDrop 2000 spectrophotometer. qRT-PCR was performed using SYBR-Green PCR kits (Takara, Shiga, Japan) and 7500 Fast Real-Time PCR System (Life Technologies, Carlsbad, CA, USA). The primers were synthesized by Tsingke biological technology. The expression level of the gene is compared with that of the housekeeping gene GAPDH. The following primers were used for RT-qPCR analysis: GAPDH, 5-ACAACTTTGGTATCGTGGAAAG-3; 5-GCCATCACGCCACAGTTTC-3 and GOT2,5′-AAGAGGGACACCATAGCAAAAAAA-3′; 5′-GCAGAACGTAAGGCTTTCCAT-3′. All experiments were performed using three complex wells.

## Scratch wound assay

$5 \times 10^5$ cells (three replicates per set) were seeded into a 6-well plate and incubated to reach confluence. Scratch the monolayer of cells using a tip and wash with serum-free medium to remove isolated cells. The cells are then cultured in complete medium containing 3% FBS. Huh7 and MHCC97H were shot after 0 h and 12 h. The wound closure area was calculated as follows: relative migration ratio (%) = (0 h wound area in experimental group - 12 h wound area in experimental group)/(0 h wound area in control group - 12 h wound area in control group) ×100.

## Statistical analysis

The Perl language is used for the data matrix and all data processing. Data analysis and visualization are performed in R (version 3.6.3) and the following packages are used for data analysis: "limma", "survival", "venn", "pheatmap", "igraph", "reshape2", "glmnet", "survminer", "Rtsne", "ggplot2", "clusterProfiler", "org. Hs.eg.db" and "enrichplot". The student *t*-test was used to identify MRGs that were differentially expressed between tumor tissues and normal tissues. When performing prognostic analysis of HCC patients,

HCC patients in the training dataset were divided into two subgroups based on the optimal cut-off value for the marker determined by the "survminer" package in R. The ratio of high-risk patients to low-risk patients in the training dataset is then applied to the validation dataset. A two-tailed $p$-value $< 0.05$ was considered statistically significant ($^*p < 0.05$, $^{**}p < 0.01$, $^{***}p < 0.001$).

## RESULT

### Identify differentially expressed genes in HCC tumor tissues

374 HCC tissues and 50 normal tissues were obtained from TCGA. By comparing the expression levels of the genes, 7,498 DEGs were found ($P < 0.01$) (Table 1). Compared with normal tissues, most genes were unrestricted in tumor tissues, with 7,104 gene expressions upregulated and 394 gene expressions downregulated in HCC tissues (Figs. 2A–2B). Among them, there were 286 metabolism-related DEGs, 228 MRGs were up-regulated, and 58 MRGs were downregulated (Figs. 2C–2D). GO analysis showed that the small molecule catabolic process, $\alpha$-amino acid metabolism process, and nucleoside phosphate biosynthesis process of HCC tumor tissue received significant changes (Fig. 2E). The results of KEGG enrichment analysis showed that the signaling pathways of purine metabolism, nucleotide metabolism, carbon metabolism, and biosynthesis of cofactors were significantly altered in tumor tissues of HCC patients (Fig. 2F). Collectively, these results suggested that the level of metabolism in tumor tissues in HCC patients is significantly different than in normal tissues.

### Analysis of the differential expression and prognostic relationship of DEGs in HCC based on TCGA database

By using univariate COX regression analysis, we found that 79 DEGs in tumor tissues of HCC patients were associated with the prognosis of HCC patients, of which 21 genes with high expression indicated a better prognosis and the other 58 genes were associated with a poor prognosis (Fig. 3A). Of these, 75 genes were MRGs (Fig. 3B). Among the 75 MRGs associated with the prognosis of HCC patients, 55 genes were highly expressed in tumor tissues of HCC patients, and 20 genes were poorly expressed. We used heatmaps for illustration (Fig. 3C). Then, we evaluated the protein-protein interactions of these genes, further revealing the strong interaction activity of these molecules at the protein level, as shown in Fig. 2D. Similarly, correlated networks built based on mRNA expression levels in TCGA showed a negative (blue) and positive (red) correlation between these prognostically relevant MRGs, as shown in Fig. 3E.

### Development and evaluation of metabolism-related risk score

The expression profiles of the above 75 genes were used to establish a prognostic model by LASSO Cox regression analysis. Based on the optimal value of $\lambda$, 14 genes that contribute the most to the prognosis of HCC patients were identified (Table 2). Among these genes, except for ACADS, GOT2, ADH4, and CSAD, high expression of the other 10 genes was associated with a poor prognosis in HCC patients ($P < 0.05$). The risk score is calculated as follows: risk score $=$ (0.0875 * DLAT expression) + (0.2953 * SEPHS1 expression)

**Table 1  DEGs between HCC tissue and normal tissue.** 374 HCC tissues and 50 normal tissues obtained from TCGA. By comparing the expression levels of the genes, 7,498 DEGs were found ($P < 0.01$).

| Gene | conMean | treatMean | logFC | pValue | fdr |
|---|---|---|---|---|---|
| IL4I1 | 0.408676109 | 1.900420927 | 2.217289189 | 4.63E−08 | 8.22E−08 |
| AKR1C1 | 27.5295606 | 64.19352837 | 1.221446277 | 0.000384398 | 0.000512757 |
| PMM2 | 0.961974242 | 1.340717505 | 0.478935118 | 0.000139748 | 0.000194667 |
| SMS | 10.69932606 | 20.11237342 | 0.910563416 | 7.48E−17 | 2.80E−16 |
| DGAT2 | 26.4610131 | 19.77988279 | −0.419834421 | 2.00E−05 | 3.01E−05 |
| IDO1 | 0.444306793 | 1.213901216 | 1.450022921 | 0.000360873 | 0.000483084 |
| ACSL4 | 7.321489016 | 46.1219159 | 2.65524345 | 3.57E−12 | 9.03E−12 |
| NAGK | 2.34741402 | 3.529596543 | 0.588430966 | 4.54E−10 | 9.44E−10 |
| GSTM4 | 3.97687114 | 6.010324011 | 0.595808951 | 0.003231848 | 0.00404651 |
| ARSA | 13.84392328 | 23.18197984 | 0.743750933 | 1.06E−09 | 2.14E−09 |
| ACSL5 | 31.7718056 | 31.6335848 | −0.006290026 | 0.000220461 | 0.000300448 |
| ACSL1 | 181.0926394 | 77.88915995 | −1.217233445 | 8.80E−18 | 3.80E−17 |
| DUT | 3.95256818 | 10.54330863 | 1.415465421 | 3.56E−24 | 5.49E−23 |
| G6PD | 1.311714874 | 13.58214647 | 3.372185434 | 6.03E−25 | 1.26E−23 |
| PCK1 | 310.8995322 | 98.70077197 | −1.655315172 | 1.33E−19 | 7.38E−19 |
| CA13 | 0.64421103 | 0.599896967 | −0.102818626 | 0.003296604 | 0.004120755 |
| ME2 | 1.285344598 | 2.315627387 | 0.84924793 | 4.85E−09 | 9.30E−09 |
| PDE3B | 2.727241112 | 2.52907411 | −0.108832939 | 0.033469155 | 0.038286685 |
| PFKP | 0.971925658 | 4.860376484 | 2.322150197 | 1.35E−06 | 2.23E−06 |
| POLE2 | 0.325707436 | 1.639062106 | 2.331221958 | 5.74E−23 | 6.11E−22 |
| NNT | 17.79027552 | 16.39284085 | −0.118022962 | 0.005707158 | 0.006949846 |
| PAFAH1B1 | 5.69548084 | 6.833324791 | 0.262770054 | 0.016430951 | 0.019203356 |
| GK | 6.33197458 | 5.984056725 | −0.081531613 | 0.009607045 | 0.011531509 |
| ASPA | 1.222286356 | 0.511516308 | −1.256730176 | 1.83E−17 | 7.57E−17 |
| AMDHD1 | 41.6883632 | 23.51702173 | −0.825939365 | 1.15E−12 | 3.01E−12 |
| ASNS | 0.500573577 | 2.570302345 | 2.36028403 | 2.43E−09 | 4.77E−09 |
| PDE6D | 2.1416811 | 4.452477304 | 1.055864581 | 5.61E−23 | 6.05E−22 |
| FMO5 | 45.5107514 | 42.94005501 | −0.083883366 | 0.014548154 | 0.017082202 |
| SRM | 12.1914571 | 32.5068222 | 1.414871963 | 2.34E−23 | 2.76E−22 |
| HCCS | 5.76547254 | 8.626777767 | 0.581382935 | 3.21E−10 | 6.81E−10 |
| POLR1B | 2.287206532 | 3.072608206 | 0.425877173 | 0.000107178 | 0.000151534 |
| ECI1 | 28.0101036 | 36.95316634 | 0.399750668 | 0.001444436 | 0.001845261 |
| PIP5K1A | 3.47425607 | 7.244620632 | 1.060206053 | 1.21E−14 | 3.84E−14 |
| LYPLA1 | 8.63802894 | 13.72384588 | 0.667910775 | 2.84E−09 | 5.50E−09 |
| CA4 | 0.042076729 | 0.569382669 | 3.758304113 | 5.34E−07 | 8.99E−07 |
| ACY3 | 17.17975988 | 15.21388658 | −0.175321118 | 0.001858912 | 0.002358787 |
| FAH | 31.8209192 | 23.57034487 | −0.433002643 | 1.15E−08 | 2.12E−08 |
| CYP2A7 | 31.69290318 | 25.23152162 | −0.328932609 | 2.65E−11 | 6.26E−11 |
| MTR | 1.27380779 | 3.363203 | 1.400688261 | 2.77E−16 | 9.85E−16 |
| NANS | 4.36062666 | 7.892830021 | 0.856007202 | 7.18E−17 | 2.70E−16 |
| LPCAT4 | 0.587157698 | 1.661035346 | 1.500262836 | 1.57E−10 | 3.47E−10 |

| Gene | conMean | treatMean | logFC | *p*Value | fdr |
|---|---|---|---|---|---|
| LPL | 0.141580397 | 1.102972743 | 2.961703712 | 1.86E−21 | 1.42E−20 |
| CYP2B6 | 126.8640398 | 29.24442074 | −2.11704987 | 1.14E−20 | 7.23E−20 |
| AOC3 | 3.922423082 | 3.742339892 | −0.067804562 | 0.003389248 | 0.004229557 |
| WARS2 | 1.72666976 | 3.099357597 | 0.843977038 | 2.58E−16 | 9.22E−16 |
| AMPD3 | 0.254917141 | 0.543059142 | 1.091080935 | 7.02E−05 | 0.000101288 |
| PCCA | 10.22722354 | 8.319849703 | −0.297785167 | 1.59E−06 | 2.60E−06 |
| BLVRB | 72.1631298 | 95.84324968 | 0.409414914 | 0.009920099 | 0.011850751 |
| NT5E | 12.01055636 | 10.74436329 | −0.16072299 | 0.000622658 | 0.000813334 |
| PSPH | 1.983490216 | 9.025365702 | 2.185944109 | 3.94E−24 | 5.83E−23 |
| PIK3C2B | 0.775023298 | 2.386662492 | 1.622682978 | 2.26E−18 | 1.05E−17 |
| LTA4H | 4.31542466 | 7.329835003 | 0.764278188 | 1.35E−18 | 6.39E−18 |
| UGT1A1 | 27.79325188 | 16.52163987 | −0.750377754 | 8.92E−09 | 1.66E−08 |
| CYP2C18 | 28.99648586 | 18.82865448 | −0.622948161 | 2.50E−09 | 4.88E−09 |
| ALDH4A1 | 101.0099032 | 72.05610849 | −0.487304101 | 1.35E−08 | 2.47E−08 |
| CYP2C9 | 287.583338 | 118.3820532 | −1.280529708 | 2.02E−17 | 8.19E−17 |
| SGMS1 | 3.20430224 | 4.119475713 | 0.362450503 | 0.006877689 | 0.008321563 |
| IMPA2 | 7.37347282 | 9.371278293 | 0.345901581 | 0.018040114 | 0.021051447 |
| GUK1 | 19.8499378 | 43.05143148 | 1.116926722 | 8.60E−22 | 7.38E−21 |
| POLE4 | 7.38593728 | 10.66305657 | 0.52976813 | 0.000190569 | 0.000261125 |
| DLAT | 3.66031604 | 7.052317534 | 0.946129215 | 1.16E−10 | 2.61E−10 |
| PDHA1 | 13.28151162 | 19.02741134 | 0.518659943 | 7.54E−11 | 1.70E−10 |
| CYB5R3 | 30.237305 | 43.89137451 | 0.53760789 | 2.01E−07 | 3.47E−07 |
| AGPAT2 | 87.005614 | 74.04673086 | −0.232672451 | 3.76E−05 | 5.54E−05 |
| OXCT1 | 0.275234787 | 1.493923329 | 2.440371378 | 0.000229146 | 0.000311721 |
| CDIPT | 20.9301402 | 32.18929631 | 0.620999063 | 1.08E−11 | 2.66E−11 |
| CA5A | 2.3876375 | 2.073194093 | −0.20372863 | 0.006409608 | 0.007780152 |
| PLA2G12B | 27.9585252 | 35.56349409 | 0.347108814 | 0.0050749 | 0.006189902 |
| CHST11 | 0.731928728 | 1.923006546 | 1.393588597 | 0.008023001 | 0.009645487 |
| POLR2G | 8.5867008 | 20.36548606 | 1.245950419 | 3.97E−26 | 1.43E−24 |
| SORD | 41.8884452 | 30.0139334 | −0.480919936 | 2.22E−07 | 3.83E−07 |
| UROS | 3.93219928 | 6.103738102 | 0.634356623 | 8.59E−11 | 1.94E−10 |
| HAL | 26.70119602 | 23.56047414 | −0.180535793 | 0.001882257 | 0.002384403 |
| UPRT | 1.800134228 | 2.776221087 | 0.625017977 | 1.73E−11 | 4.18E−11 |
| ENTPD6 | 4.89225452 | 13.77275387 | 1.493245689 | 7.12E−25 | 1.41E−23 |
| SPHK1 | 0.741709572 | 5.347719945 | 2.849997622 | 8.75E−05 | 0.000124871 |
| CANT1 | 4.13244666 | 10.66555898 | 1.367891476 | 6.59E−25 | 1.34E−23 |
| ACADL | 4.060608256 | 2.250819133 | −0.851245719 | 6.55E−12 | 1.63E−11 |
| MIF | 18.5365111 | 45.69842886 | 1.301774837 | 2.78E−13 | 7.93E−13 |
| UCK1 | 8.54689772 | 14.03707748 | 0.715769834 | 3.83E−13 | 1.08E−12 |
| GALT | 12.9575935 | 16.26293456 | 0.327789803 | 0.004091075 | 0.005046996 |
| ACSM1 | 3.973052599 | 15.02401635 | 1.91895074 | 0.000302397 | 0.000406969 |
| ALDH1A3 | 0.565172746 | 0.487798622 | −0.212406216 | 8.54E−06 | 1.33E−05 |
| POLR2K | 10.2636964 | 29.85350878 | 1.540350105 | 2.97E−28 | 3.93E−26 |

**Table 1** (*continued*)

| Gene | conMean | treatMean | logFC | *p*Value | fdr |
|---|---|---|---|---|---|
| PCK2 | 145.7806962 | 90.77617087 | −0.683414157 | 6.45E−11 | 1.47E−10 |
| LPCAT1 | 1.635662076 | 6.94731366 | 2.086580512 | 1.78E−16 | 6.50E−16 |
| NAA80 | 1.86636393 | 4.187413568 | 1.165829082 | 1.23E−18 | 5.83E−18 |
| ADCY9 | 2.877553656 | 4.347904755 | 0.595477507 | 3.01E−06 | 4.85E−06 |
| GMPPA | 5.56356326 | 11.56588251 | 1.055794274 | 1.45E−24 | 2.61E−23 |
| CRLS1 | 14.10303008 | 21.11234898 | 0.58208194 | 0.000158942 | 0.000219782 |
| SEPHS1 | 7.92983666 | 13.77461218 | 0.796648647 | 1.75E−20 | 1.06E−19 |
| CYP2C19 | 3.085339415 | 0.382142328 | −3.013247236 | 3.19E−17 | 1.25E−16 |
| KDSR | 7.4475128 | 6.084439599 | −0.291634307 | 6.26E−05 | 9.06E−05 |
| PFKFB2 | 0.382136054 | 1.420710904 | 1.89445473 | 6.30E−20 | 3.60E−19 |
| POLR3H | 2.7771828 | 4.797098075 | 0.788539794 | 1.59E−16 | 5.83E−16 |
| CYP4F3 | 37.5404 | 30.65832969 | −0.292164924 | 1.27E−05 | 1.94E−05 |
| UXS1 | 1.5551954 | 4.725752008 | 1.603448067 | 1.67E−26 | 7.87E−25 |
| POLA2 | 0.664501946 | 2.587714342 | 1.961333037 | 8.15E−26 | 2.28E−24 |
| NT5C | 5.03591866 | 11.91640459 | 1.242622127 | 8.44E−19 | 4.19E−18 |
| BDH1 | 21.2007566 | 16.85562386 | −0.330885726 | 0.000126492 | 0.000176528 |
| LIPC | 26.95726792 | 20.48967436 | −0.395777233 | 3.01E−06 | 4.85E−06 |
| PLA2G5 | 1.153261896 | 0.89094941 | −0.372304753 | 0.000109361 | 0.000153757 |
| PLA2G6 | 0.58048134 | 2.296062017 | 1.983840014 | 5.46E−24 | 7.63E−23 |
| NMNAT1 | 1.26411244 | 1.805223587 | 0.514052741 | 9.85E−06 | 1.52E−05 |
| ACO2 | 12.52959346 | 20.83091178 | 0.733386383 | 3.46E−08 | 6.20E−08 |
| FTH1 | 139.0992968 | 286.4863544 | 1.042351297 | 8.93E−17 | 3.30E−16 |
| ACADS | 72.100922 | 32.05868751 | −1.169302346 | 9.35E−22 | 7.93E−21 |
| GGT5 | 7.32869966 | 4.290134866 | −0.772534241 | 1.15E−12 | 3.01E−12 |
| ALDH1A1 | 267.4815334 | 396.7508063 | 0.568793862 | 0.021467023 | 0.024858286 |
| TPMT | 20.16445966 | 18.52541508 | −0.122308879 | 0.000533541 | 0.000700562 |
| PDE5A | 0.195980374 | 0.503871718 | 1.362347291 | 1.58E−06 | 2.59E−06 |
| PRIM1 | 0.936742084 | 3.706683544 | 1.984405165 | 4.14E−21 | 2.89E−20 |
| HIBCH | 7.9563316 | 6.281329144 | −0.341033536 | 5.63E−06 | 8.82E−06 |
| RDH10 | 9.06576346 | 15.41329115 | 0.765674524 | 1.50E−06 | 2.46E−06 |
| HEMK1 | 0.65588727 | 1.173926415 | 0.8398222 | 7.25E−21 | 4.89E−20 |
| LPGAT1 | 8.05441224 | 21.31594928 | 1.404082087 | 2.17E−17 | 8.77E−17 |
| TYMS | 1.561858024 | 8.89844653 | 2.51029018 | 1.44E−20 | 8.89E−20 |
| DNMT1 | 1.168587338 | 4.380206253 | 1.906233242 | 7.50E−21 | 4.97E−20 |
| PC | 48.5013028 | 37.34714447 | −0.37702556 | 4.28E−06 | 6.78E−06 |
| POLR1A | 0.958182464 | 2.696126111 | 1.492515664 | 3.87E−26 | 1.43E−24 |
| CAT | 172.3659592 | 98.46953686 | −0.807725505 | 2.28E−14 | 6.99E−14 |
| CP | 186.4100454 | 97.23571025 | −0.938921455 | 7.95E−13 | 2.13E−12 |
| DBH | 10.78850207 | 1.804720176 | −2.5796475 | 1.78E−24 | 3.10E−23 |
| UGT2B10 | 198.5524482 | 92.80896422 | −1.097184086 | 9.18E−13 | 2.45E−12 |
| GNPDA2 | 0.319322526 | 0.662430617 | 1.052755026 | 1.31E−10 | 2.91E−10 |
| NME4 | 16.31808562 | 24.98593807 | 0.614644568 | 0.000109914 | 0.000154247 |

| Gene | conMean | treatMean | logFC | pValue | fdr |
|------|---------|-----------|-------|--------|-----|
| MLYCD | 2.48341314 | 1.547413766 | −0.682465274 | 8.55E−15 | 2.74E−14 |
| PFKFB1 | 8.5895856 | 7.825367716 | −0.134429985 | 0.002113498 | 0.002672849 |
| SULT1A2 | 9.07027648 | 6.751247492 | −0.42599242 | 1.14E−05 | 1.75E−05 |
| GALK1 | 24.967336 | 42.45332342 | 0.765835604 | 9.54E−05 | 0.000135645 |
| PHGDH | 23.51638218 | 13.43361449 | −0.807818596 | 2.77E−11 | 6.48E−11 |
| GPAT2 | 0.090898479 | 0.478071167 | 2.394897342 | 2.26E−10 | 4.91E−10 |
| OPLAH | 8.82522278 | 16.89672561 | 0.937039094 | 2.09E−10 | 4.56E−10 |
| IMPDH1 | 1.350320208 | 4.233778779 | 1.648644327 | 7.85E−07 | 1.31E−06 |
| NPR2 | 2.597858634 | 5.96065675 | 1.19814837 | 0.000471399 | 0.000621127 |
| FMO3 | 137.9187216 | 93.88484103 | −0.554854168 | 3.50E−08 | 6.27E−08 |
| AACS | 0.31328518 | 1.03790029 | 1.728119422 | 1.45E−19 | 7.96E−19 |
| NMRK1 | 6.37048312 | 5.351685995 | −0.251409317 | 8.27E−05 | 0.00011827 |
| ENTPD5 | 22.40752186 | 18.09005981 | −0.308785926 | 0.000475759 | 0.000625781 |
| MPST | 34.8617012 | 45.54922933 | 0.385783672 | 0.003609969 | 0.004497569 |
| MTMR1 | 1.628000482 | 3.071386948 | 0.915789155 | 1.92E−14 | 5.94E−14 |
| GLA | 3.24193056 | 11.36254672 | 1.809361132 | 1.20E−24 | 2.21E−23 |
| MGST3 | 5.56648038 | 10.14191067 | 0.865492151 | 1.91E−17 | 7.85E−17 |
| EPHX1 | 524.41677 | 1035.59981 | 0.981680879 | 7.88E−06 | 1.23E−05 |
| LYPLA2 | 21.932233 | 34.59496813 | 0.657509506 | 4.98E−13 | 1.38E−12 |
| CAD | 1.068638102 | 4.256222464 | 1.993800198 | 5.58E−26 | 1.76E−24 |
| NEU3 | 0.572362622 | 1.294232685 | 1.177095651 | 1.28E−17 | 5.36E−17 |
| EHHADH | 68.1731674 | 41.91216309 | −0.701835033 | 2.39E−10 | 5.18E−10 |
| CTPS2 | 1.630246474 | 3.255587356 | 0.997827751 | 8.08E−18 | 3.53E−17 |
| MTHFD2 | 0.687717227 | 1.248735294 | 0.860580299 | 0.006481558 | 0.007854858 |
| SMPD3 | 0.70683542 | 0.332301761 | −1.088880396 | 4.31E−17 | 1.65E−16 |
| RDH16 | 121.453585 | 43.35288581 | −1.48620514 | 1.45E−19 | 7.96E−19 |
| DGKZ | 1.274013736 | 3.212991644 | 1.334536397 | 1.11E−23 | 1.47E−22 |
| TPI1 | 83.9320608 | 139.4822363 | 0.73278749 | 5.71E−14 | 1.69E−13 |
| LCMT2 | 0.797014992 | 1.12892053 | 0.502265164 | 4.15E−07 | 7.04E−07 |
| TAT | 265.3094254 | 154.2425759 | −0.782474879 | 2.12E−09 | 4.19E−09 |
| HEXA | 3.18901968 | 6.698987527 | 1.070830065 | 1.23E−18 | 5.83E−18 |
| ARG1 | 240.2322 | 166.3619414 | −0.530104112 | 1.68E−08 | 3.06E−08 |
| CPT1C | 0.112497877 | 0.356503529 | 1.664018582 | 1.18E−13 | 3.43E−13 |
| MTHFD2L | 1.241255578 | 0.761231939 | −0.705392202 | 5.19E−10 | 1.08E−09 |
| MTAP | 1.15203675 | 1.583067801 | 0.458536306 | 2.94E−06 | 4.75E−06 |
| RDH5 | 4.71202044 | 2.420467525 | −0.961060059 | 6.15E−15 | 1.98E−14 |
| DNMT3L | 0.424289348 | 0.291648168 | −0.540819449 | 2.06E−05 | 3.08E−05 |
| HPRT1 | 14.17009836 | 18.06220948 | 0.35012461 | 0.009675838 | 0.011577274 |
| ASS1 | 584.625332 | 252.8417532 | −1.209277619 | 1.11E−18 | 5.38E−18 |
| ACHE | 0.686224915 | 1.858858895 | 1.437663848 | 0.032860592 | 0.037762172 |
| BUD23 | 4.074106 | 8.36906785 | 1.038583427 | 1.30E−25 | 3.27E−24 |

**Table 1** (*continued*)

| Gene | conMean | treatMean | logFC | pValue | fdr |
|------|---------|-----------|-------|--------|-----|
| GMPPB | 1.87040842 | 4.208787714 | 1.170051414 | 8.29E−23 | 8.57E−22 |
| PLCD4 | 0.125425913 | 0.438557777 | 1.805931481 | 1.23E−18 | 5.83E−18 |
| PTDSS1 | 6.74127936 | 12.91530242 | 0.937987108 | 5.06E−15 | 1.64E−14 |
| CS | 5.37878536 | 13.8183489 | 1.361232919 | 1.45E−22 | 1.42E−21 |
| PIKFYVE | 1.39647462 | 1.870835031 | 0.421892995 | 0.004941202 | 0.006046366 |
| POLD3 | 0.842582454 | 2.064203063 | 1.292695122 | 2.33E−19 | 1.24E−18 |
| AOX1 | 221.5975442 | 121.9637437 | −0.861489553 | 1.55E−11 | 3.77E−11 |
| HMGCS2 | 664.916598 | 435.8643833 | −0.609294073 | 5.83E−10 | 1.21E−09 |
| FPGT | 1.601297378 | 2.672968881 | 0.739201785 | 9.43E−13 | 2.51E−12 |
| POLR2A | 8.00200448 | 10.08390693 | 0.333621366 | 0.026768153 | 0.030902072 |
| SGPP1 | 6.7100943 | 6.026959952 | −0.154902563 | 0.004352827 | 0.005352418 |
| POLD2 | 19.6641202 | 29.83523623 | 0.60145156 | 7.12E−09 | 1.33E−08 |
| DGAT1 | 11.44228732 | 19.98911916 | 0.804839423 | 2.18E−09 | 4.30E−09 |
| PNP | 9.04160578 | 7.688707168 | −0.233837983 | 0.000414134 | 0.000550477 |
| ACADSB | 72.1887034 | 32.77565001 | −1.139148699 | 1.46E−18 | 6.86E−18 |
| UGT2B11 | 0.932454448 | 13.38406332 | 3.843339116 | 1.32E−12 | 3.46E−12 |
| HK3 | 1.755092942 | 1.065993519 | −0.719348764 | 1.59E−10 | 3.50E−10 |
| UCK2 | 0.97723134 | 4.97742167 | 2.348626575 | 2.02E−28 | 3.82E−26 |
| NPL | 2.542158828 | 4.829610362 | 0.925852632 | 0.012594123 | 0.014833951 |
| B4GALT6 | 0.297784155 | 0.661726029 | 1.151967038 | 2.51E−07 | 4.30E−07 |
| PLCB3 | 1.634257378 | 3.846683794 | 1.234980031 | 1.86E−21 | 1.42E−20 |
| DLD | 11.88321106 | 14.92935626 | 0.329227228 | 0.002449057 | 0.003086875 |
| POLR1E | 7.44986644 | 6.602367032 | −0.17423122 | 0.003118202 | 0.003917209 |
| POLR3K | 2.48922156 | 5.518207323 | 1.148505014 | 2.44E−21 | 1.81E−20 |
| OGDHL | 26.5042672 | 14.50832449 | −0.869343735 | 4.39E−13 | 1.23E−12 |
| ALAS1 | 159.56769 | 100.1414548 | −0.672129238 | 1.81E−09 | 3.59E−09 |
| POLR3D | 0.963096128 | 1.733721133 | 0.848120154 | 5.64E−08 | 9.99E−08 |
| INMT | 9.65272184 | 2.507403196 | −1.944741802 | 7.33E−21 | 4.90E−20 |
| GUSB | 30.392177 | 38.53529635 | 0.342480467 | 0.002655896 | 0.003342003 |
| CPS1 | 205.3781916 | 136.0158919 | −0.594507771 | 2.45E−07 | 4.19E−07 |
| METTL6 | 0.430309778 | 1.185576642 | 1.462141401 | 6.71E−28 | 5.07E−26 |
| GOT2 | 87.1599586 | 65.06480682 | −0.421788102 | 2.46E−09 | 4.82E−09 |
| PTGES | 0.297396069 | 3.200945654 | 3.428040699 | 0.014797686 | 0.017348219 |
| AGPS | 3.7115833 | 5.295453451 | 0.51271948 | 2.80E−05 | 4.14E−05 |
| POLR1D | 4.90762592 | 8.660998203 | 0.819508024 | 2.72E−18 | 1.25E−17 |
| ACSL3 | 6.99338192 | 11.86570804 | 0.76273599 | 6.24E−09 | 1.17E−08 |
| DHRS4 | 12.25678654 | 11.53249242 | −0.087876441 | 0.020310735 | 0.023555461 |
| HYI | 3.01594442 | 5.084056814 | 0.753370311 | 2.55E−11 | 6.04E−11 |
| GLUD1 | 180.0245172 | 133.4265423 | −0.43214771 | 8.27E−10 | 1.69E−09 |
| POLD4 | 8.05781742 | 12.54161247 | 0.638261825 | 5.34E−09 | 1.01E−08 |
| PPOX | 1.196744276 | 4.192724463 | 1.808773117 | 1.44E−29 | 6.98E−27 |
| ADA | 0.943392724 | 2.67245149 | 1.502233382 | 1.19E−17 | 5.01E−17 |
| HMOX1 | 45.83893488 | 27.73183896 | −0.725030234 | 9.52E−09 | 1.77E−08 |

**Table 1** (*continued*)

| Gene | conMean | treatMean | logFC | *p*Value | fdr |
|------|---------|-----------|-------|----------|-----|
| DMGDH | 31.7591144 | 15.47932348 | −1.036828263 | 5.46E−16 | 1.88E−15 |
| AHCY | 44.0853498 | 69.89104425 | 0.664808295 | 2.94E−08 | 5.30E−08 |
| PRIM2 | 0.502543684 | 1.859031108 | 1.887229998 | 4.42E−27 | 2.57E−25 |
| ACADM | 31.879362 | 20.11711322 | −0.664199462 | 2.07E−11 | 4.96E−11 |
| PIK3CB | 1.966956368 | 3.395752372 | 0.787766302 | 5.26E−13 | 1.44E−12 |
| PRUNE1 | 3.4628476 | 9.611363235 | 1.472782175 | 5.02E−23 | 5.49E−22 |
| CHKB | 0.422492212 | 1.182076973 | 1.484327329 | 9.12E−19 | 4.50E−18 |
| MGLL | 13.76947858 | 12.46297017 | −0.143825997 | 0.009675841 | 0.011577274 |
| OCRL | 1.958109446 | 5.65422446 | 1.529867751 | 1.85E−24 | 3.10E−23 |
| LDHD | 55.065139 | 32.56811545 | −0.757679016 | 6.24E−13 | 1.70E−12 |
| HMBS | 2.84492594 | 6.077665142 | 1.095126093 | 4.39E−23 | 4.87E−22 |
| DPYD | 9.11150098 | 7.318402323 | −0.316160007 | 6.13E−05 | 8.89E−05 |
| ABAT | 53.955323 | 26.73360559 | −1.013110868 | 3.57E−16 | 1.26E−15 |
| ADH6 | 89.808434 | 46.97759442 | −0.934878097 | 6.33E−14 | 1.86E−13 |
| PDE7B | 0.930007826 | 0.456690251 | −1.026026862 | 1.31E−15 | 4.37E−15 |
| PISD | 2.83145052 | 4.389637725 | 0.632560561 | 7.78E−10 | 1.60E−09 |
| PIP5K1C | 1.467138022 | 3.719134168 | 1.341962194 | 3.88E−23 | 4.44E−22 |
| ALDH3A1 | 0.752936723 | 44.07999412 | 5.871451597 | 0.000343499 | 0.000460643 |
| GBA | 5.56522966 | 22.70087955 | 2.028235066 | 3.64E−28 | 3.93E−26 |
| GALK2 | 1.588663302 | 2.448799751 | 0.624261408 | 4.17E−08 | 7.43E−08 |
| MAOB | 92.0239682 | 75.53398313 | −0.284883802 | 2.02E−06 | 3.29E−06 |
| ACAT1 | 82.9427028 | 47.2412729 | −0.812067222 | 2.12E−16 | 7.66E−16 |
| UGDH | 22.73636822 | 41.03832833 | 0.85197014 | 0.000206002 | 0.00028176 |
| EPHX2 | 46.3273598 | 24.00327051 | −0.948633475 | 3.57E−17 | 1.37E−16 |
| CERK | 5.53242282 | 11.22309283 | 1.02048698 | 1.28E−14 | 4.03E−14 |
| TAZ | 1.723546396 | 5.52082373 | 1.679503404 | 7.52E−29 | 1.89E−26 |
| PGD | 20.44094368 | 39.89972749 | 0.964917092 | 3.83E−09 | 7.38E−09 |
| ACO1 | 20.96810206 | 19.6283929 | −0.095254226 | 0.00010772 | 0.000151732 |
| AMDHD2 | 2.031742526 | 4.730317108 | 1.219219314 | 1.28E−20 | 7.99E−20 |
| PDE6G | 0.86451153 | 0.639854258 | −0.434141871 | 5.70E−05 | 8.28E−05 |
| CYP4A22 | 60.5277696 | 19.74899941 | −1.61581763 | 1.17E−20 | 7.34E−20 |
| SDS | 194.0116684 | 163.218774 | −0.249336412 | 4.38E−06 | 6.93E−06 |
| FMO4 | 10.20612596 | 9.414288939 | −0.116511315 | 0.003193554 | 0.004005204 |
| GPAT3 | 5.552175014 | 4.162315257 | −0.415666805 | 0.012249465 | 0.014473155 |
| PFKL | 12.61516374 | 19.37322019 | 0.618904845 | 2.04E−10 | 4.46E−10 |
| HAO1 | 172.769988 | 92.22580563 | −0.905610236 | 3.35E−17 | 1.30E−16 |
| CYP4F2 | 60.3235252 | 29.46836538 | −1.033553704 | 6.21E−15 | 1.99E−14 |
| FTCD | 155.5260496 | 86.5196214 | −0.846056985 | 6.71E−13 | 1.81E−12 |
| HPD | 605.159282 | 532.9277054 | −0.183375083 | 0.000107178 | 0.000151534 |
| PDE4B | 1.062459026 | 0.963981326 | −0.1403301 | 0.0009672 | 0.00125686 |
| PRDX6 | 171.8414344 | 221.4824048 | 0.36611415 | 0.001355076 | 0.001736982 |
| SUCLG1 | 21.1592364 | 27.17118061 | 0.360789689 | 0.000143973 | 0.000200183 |

| Gene | conMean | treatMean | logFC | pValue | fdr |
| --- | --- | --- | --- | --- | --- |
| PIK3C2G | 2.247678282 | 1.473184304 | −0.609497621 | 2.52E−08 | 4.56E−08 |
| PGM3 | 4.06042848 | 5.511269867 | 0.440752795 | 0.00140208 | 0.001794187 |
| GPAT4 | 5.23151944 | 9.836442457 | 0.910906608 | 1.77E−12 | 4.55E−12 |
| CHDH | 7.9646379 | 11.15826518 | 0.486432065 | 9.46E−06 | 1.46E−05 |
| BLVRA | 3.276705816 | 12.07960422 | 1.882255129 | 5.93E−14 | 1.75E−13 |
| MARS2 | 1.96582365 | 2.821305799 | 0.52122914 | 4.85E−05 | 7.09E−05 |
| PSAT1 | 58.4287532 | 37.24670641 | −0.649565645 | 3.01E−10 | 6.41E−10 |
| ALDH5A1 | 25.002095 | 21.66769979 | −0.20650298 | 0.00010772 | 0.000151732 |
| GGT7 | 3.07695892 | 5.573133026 | 0.856983407 | 2.99E−11 | 6.94E−11 |
| AGPAT4 | 0.136465682 | 0.57271225 | 2.069272273 | 9.51E−10 | 1.93E−09 |
| ALDH9A1 | 48.6317048 | 40.3139759 | −0.270617093 | 2.24E−06 | 3.64E−06 |
| PI4KB | 4.0879725 | 11.41793196 | 1.481843974 | 1.67E−26 | 7.87E−25 |
| GALE | 10.83432624 | 18.55155752 | 0.775930877 | 2.02E−11 | 4.85E−11 |
| PLD1 | 2.33434292 | 2.279957658 | −0.03400948 | 0.026683904 | 0.030851987 |
| CYP2U1 | 1.250805474 | 1.14419921 | −0.128519184 | 0.012594123 | 0.014833951 |
| ENOPH1 | 5.50486998 | 11.72488047 | 1.090792821 | 2.78E−21 | 2.03E−20 |
| AGXT | 394.753526 | 250.6042388 | −0.655541335 | 4.40E−10 | 9.18E−10 |
| PLCB2 | 1.160730889 | 1.104038541 | −0.072242991 | 0.040462466 | 0.046007774 |
| CYP1B1 | 1.337915968 | 5.94969667 | 2.152828612 | 0.006928705 | 0.008369875 |
| NT5C2 | 1.540448554 | 2.299890391 | 0.578214605 | 0.001724337 | 0.002195404 |
| NME6 | 1.050427416 | 2.501002343 | 1.251529933 | 3.81E−27 | 2.40E−25 |
| HMGCL | 40.4827466 | 24.59046419 | −0.719208206 | 2.52E−16 | 9.07E−16 |
| SAT1 | 121.6407268 | 113.7395081 | −0.096892871 | 0.033469155 | 0.038286685 |
| SUCLG2 | 54.220537 | 39.39971703 | −0.460654133 | 3.94E−10 | 8.24E−10 |
| ADH1A | 388.209718 | 164.8505374 | −1.235677643 | 5.65E−17 | 2.13E−16 |
| CTH | 37.1185504 | 29.77105419 | −0.318230061 | 1.40E−08 | 2.56E−08 |
| CYP3A4 | 765.778506 | 344.3796666 | −1.152927204 | 1.69E−15 | 5.61E−15 |
| ECHS1 | 468.98201 | 301.2351756 | −0.638642337 | 5.40E−13 | 1.48E−12 |
| PGM1 | 59.3355864 | 38.43717944 | −0.626395144 | 1.11E−13 | 3.24E−13 |
| POLD1 | 1.184827396 | 4.730074491 | 1.997185999 | 1.56E−26 | 7.87E−25 |
| NAMPT | 30.29449552 | 16.41230958 | −0.884277408 | 2.28E−07 | 3.92E−07 |
| PFKM | 0.866943006 | 2.25856606 | 1.381398054 | 3.07E−06 | 4.93E−06 |
| SCLY | 0.20643543 | 0.46320077 | 1.165947056 | 1.86E−14 | 5.79E−14 |
| CHKA | 3.36662532 | 11.5937026 | 1.78396631 | 5.27E−22 | 4.74E−21 |
| GDA | 6.26289082 | 5.902956593 | −0.085390996 | 0.003981116 | 0.004927447 |
| GPX2 | 106.9847132 | 267.7036539 | 1.323232162 | 0.010097916 | 0.012044118 |
| PAPSS2 | 20.1801266 | 15.74261221 | −0.358260274 | 1.30E−05 | 1.98E−05 |
| PLCG1 | 1.260780372 | 3.995510695 | 1.664062936 | 1.77E−21 | 1.39E−20 |
| PIP4K2C | 4.43018188 | 8.973248829 | 1.018264488 | 9.69E−18 | 4.13E−17 |
| EARS2 | 3.1501315 | 5.085392201 | 0.690946989 | 1.33E−11 | 3.27E−11 |
| GSS | 17.1787072 | 29.50461679 | 0.780319251 | 4.06E−18 | 1.85E−17 |
| AK3 | 31.4693146 | 24.77009647 | −0.34534627 | 3.86E−08 | 6.90E−08 |

| Gene | conMean | treatMean | logFC | *p*Value | fdr |
|---|---|---|---|---|---|
| APRT | 28.6023808 | 54.41106794 | 0.927764906 | 3.86E−14 | 1.16E−13 |
| FMO1 | 0.209436746 | 1.621167947 | 2.952447066 | 0.003981116 | 0.004927447 |
| DHRS4L2 | 11.34542994 | 10.55728929 | −0.103871829 | 0.00698006 | 0.008405017 |
| AKR1C3 | 17.68298576 | 83.41396337 | 2.237927015 | 1.18E−23 | 1.54E−22 |
| GNPAT | 8.1486725 | 23.37881262 | 1.520564705 | 1.98E−27 | 1.36E−25 |
| UROD | 21.544076 | 30.24876272 | 0.489584908 | 5.04E−09 | 9.60E−09 |
| CBR1 | 58.780441 | 99.21114632 | 0.755166033 | 0.001746125 | 0.002219401 |
| SMPD4 | 2.28095394 | 5.437838743 | 1.253396055 | 2.14E−22 | 2.02E−21 |
| GPX7 | 1.50203363 | 4.344910673 | 1.532409402 | 0.000302397 | 0.000406969 |
| GSTO2 | 0.303990672 | 0.880769726 | 1.534737826 | 0.001201774 | 0.001545723 |
| GSTK1 | 45.1586356 | 63.78578441 | 0.498233036 | 8.94E−08 | 1.57E−07 |
| CMAS | 15.22241776 | 20.10836615 | 0.401598346 | 5.00E−05 | 7.30E−05 |
| PGP | 2.46275828 | 8.344557131 | 1.76056045 | 1.37E−25 | 3.34E−24 |
| LRAT | 1.094300194 | 0.233448683 | −2.228831208 | 9.54E−25 | 1.80E−23 |
| UCKL1 | 4.9052273 | 11.12656123 | 1.181615888 | 1.80E−23 | 2.23E−22 |
| PIP4K2B | 3.30158812 | 7.811014925 | 1.242349865 | 1.56E−23 | 1.96E−22 |
| ACOT12 | 21.45356354 | 13.71772584 | −0.645175979 | 9.38E−09 | 1.75E−08 |
| GSR | 15.11395882 | 25.70655358 | 0.766254607 | 4.11E−05 | 6.04E−05 |
| CPT2 | 25.1462894 | 16.46242324 | −0.611168817 | 2.48E−12 | 6.29E−12 |
| PLD2 | 1.824123198 | 3.033992898 | 0.734014539 | 8.11E−09 | 1.52E−08 |
| POLR3GL | 12.67931058 | 19.32978751 | 0.608349475 | 1.82E−09 | 3.61E−09 |
| GGCT | 7.3906693 | 16.97966798 | 1.200031323 | 1.92E−23 | 2.34E−22 |
| CYB5R1 | 6.89941468 | 17.42151749 | 1.336324415 | 1.52E−25 | 3.59E−24 |
| PDE2A | 2.178131344 | 1.126729193 | −0.950950145 | 1.15E−13 | 3.35E−13 |
| OGDH | 13.79518708 | 24.98405926 | 0.856842875 | 4.59E−13 | 1.28E−12 |
| ENPP1 | 9.1646637 | 7.547183961 | −0.280143502 | 1.15E−05 | 1.76E−05 |
| COMT | 30.706629 | 25.72272779 | −0.255506498 | 2.65E−05 | 3.94E−05 |
| QPRT | 43.8780206 | 54.08350685 | 0.301690256 | 0.04267486 | 0.048305126 |
| AGMAT | 32.4994148 | 25.32091879 | −0.360083985 | 2.71E−05 | 4.02E−05 |
| GRHPR | 62.909347 | 40.6577625 | −0.629743565 | 9.98E−12 | 2.48E−11 |
| ACER2 | 0.36464292 | 0.323416101 | −0.173092878 | 0.012247144 | 0.014473155 |
| FLAD1 | 4.34967458 | 13.71639311 | 1.656921782 | 1.85E−29 | 6.98E−27 |
| PYCR3 | 2.93010406 | 9.072809425 | 1.630597454 | 9.42E−25 | 1.80E−23 |
| HEXB | 14.85885542 | 30.48633278 | 1.036839629 | 3.86E−21 | 2.72E−20 |
| NAT1 | 2.219190636 | 1.399346689 | −0.66528017 | 2.09E−11 | 4.98E−11 |
| GART | 3.11647116 | 5.301358952 | 0.766448866 | 3.40E−14 | 1.03E−13 |
| PLCE1 | 0.212974455 | 0.608776786 | 1.515232948 | 9.85E−19 | 4.83E−18 |
| PLCD3 | 0.266926144 | 1.228215767 | 2.202051505 | 8.51E−16 | 2.88E−15 |
| CYP3A43 | 2.620158394 | 1.416590761 | −0.887230991 | 1.35E−12 | 3.50E−12 |
| ETNK2 | 33.401069 | 32.2062868 | −0.052551941 | 0.004773988 | 0.005851236 |
| CDS2 | 2.7498919 | 4.632668795 | 0.752468636 | 1.26E−14 | 3.97E−14 |
| NME2 | 6.86305604 | 19.23153189 | 1.486550645 | 1.71E−21 | 1.36E−20 |

**Table 1** (*continued*)

| Gene | conMean | treatMean | logFC | *p*Value | fdr |
|---|---|---|---|---|---|
| IMPDH2 | 12.91924434 | 36.52956009 | 1.499542692 | 3.89E−24 | 5.83E−23 |
| OTC | 88.002804 | 58.70658403 | −0.58402718 | 6.74E−07 | 1.13E−06 |
| PIK3C2A | 2.259323538 | 3.277646011 | 0.536769169 | 3.36E−05 | 4.96E−05 |
| CYP4A11 | 233.0618168 | 72.59752175 | −1.682720458 | 1.82E−24 | 3.10E−23 |
| ACSS1 | 0.750883248 | 2.273225868 | 1.598080527 | 1.27E−05 | 1.94E−05 |
| POLE | 1.3055207 | 2.220828061 | 0.766472369 | 6.30E−10 | 1.30E−09 |
| ACP1 | 11.20816652 | 18.54387434 | 0.7263924 | 8.05E−21 | 5.19E−20 |
| ACMSD | 26.174961 | 21.07948543 | −0.312347738 | 4.05E−05 | 5.96E−05 |
| CYP2A6 | 536.9454174 | 278.4392503 | −0.947412844 | 2.29E−11 | 5.43E−11 |
| SHMT1 | 52.7147148 | 32.16605069 | −0.712666918 | 1.45E−12 | 3.75E−12 |
| CEL | 0.033783259 | 0.531241933 | 3.974988605 | 3.70E−16 | 1.30E−15 |
| MIOX | 0.025307544 | 0.730525513 | 4.851295268 | 1.25E−11 | 3.07E−11 |
| PLA2G7 | 1.73344392 | 4.023914912 | 1.214958635 | 0.000584406 | 0.00076469 |
| GNPNAT1 | 13.14352562 | 9.498633602 | −0.468560417 | 5.97E−09 | 1.12E−08 |
| PNPT1 | 2.55535368 | 4.267066066 | 0.739721463 | 3.15E−14 | 9.59E−14 |
| INPP5K | 3.03390274 | 4.147665679 | 0.451124774 | 4.84E−07 | 8.16E−07 |
| ADK | 19.8979868 | 13.99084463 | −0.508139411 | 1.05E−11 | 2.60E−11 |
| CA12 | 0.325978387 | 3.7360177 | 3.518653069 | 1.48E−09 | 2.96E−09 |
| NMNAT3 | 0.695570616 | 1.378986438 | 0.987339375 | 1.16E−09 | 2.33E−09 |
| KMO | 8.11238004 | 3.338421404 | −1.280959164 | 8.62E−18 | 3.74E−17 |
| ARG2 | 0.49592443 | 2.010793536 | 2.019572755 | 7.62E−05 | 0.000109157 |
| GNPDA1 | 2.73301806 | 7.292774088 | 1.41597271 | 3.86E−20 | 2.26E−19 |
| MAT2A | 11.17323804 | 17.09991339 | 0.613941674 | 0.000105036 | 0.000149064 |
| AGK | 1.495999866 | 2.767032531 | 0.887229559 | 4.65E−16 | 1.62E−15 |
| UROC1 | 40.84584802 | 11.96254955 | −1.771664535 | 3.22E−19 | 1.67E−18 |
| ADH4 | 804.750736 | 228.0260374 | −1.819343421 | 8.53E−21 | 5.46E−20 |
| UGT2B7 | 279.779916 | 115.0979264 | −1.281430561 | 6.96E−18 | 3.07E−17 |
| PAH | 116.2219842 | 94.99517702 | −0.290956817 | 0.000150537 | 0.000208925 |
| ALOX15B | 0.142066661 | 1.817887709 | 3.677623143 | 7.72E−07 | 1.29E−06 |
| GMPR2 | 7.85884262 | 12.66631626 | 0.688608242 | 1.09E−18 | 5.30E−18 |
| GAA | 20.41744244 | 34.96541419 | 0.776126435 | 1.55E−11 | 3.77E−11 |
| APIP | 1.640616216 | 3.661128344 | 1.158050556 | 3.30E−22 | 3.00E−21 |
| PFAS | 1.153738786 | 2.725413061 | 1.240158275 | 5.61E−19 | 2.82E−18 |
| IMPA1 | 4.43702154 | 7.935114115 | 0.838659414 | 2.13E−12 | 5.44E−12 |
| NAT2 | 25.52141146 | 4.833929648 | −2.400439738 | 1.20E−25 | 3.13E−24 |
| SMPD2 | 1.62738852 | 3.829329681 | 1.234533154 | 1.20E−19 | 6.72E−19 |
| SUCLG2P2 | 0.361432093 | 0.260115564 | −0.474571886 | 7.34E−05 | 0.000105697 |
| SYNJ1 | 0.648474406 | 0.91861337 | 0.502408146 | 0.000217285 | 0.000296654 |
| PAICS | 11.60426494 | 16.32540539 | 0.492463677 | 1.22E−08 | 2.24E−08 |
| TK1 | 2.468539172 | 20.42635341 | 3.048702228 | 5.30E−26 | 1.74E−24 |
| AFMID | 18.4721284 | 24.5251126 | 0.408909654 | 0.000233607 | 0.000317219 |
| ALAD | 38.4233746 | 37.10296135 | −0.050449893 | 0.01920916 | 0.022381043 |

| Gene | conMean | treatMean | logFC | *p*Value | fdr |
|------|---------|-----------|-------|----------|-----|
| NT5C3A | 1.954837404 | 4.189451043 | 1.099712601 | 2.24E−17 | 9.01E−17 |
| GSTZ1 | 11.81957816 | 4.287539533 | −1.462956668 | 2.84E−21 | 2.06E−20 |
| PGM2L1 | 0.22568328 | 0.446572794 | 0.984595823 | 0.001355076 | 0.001736982 |
| DAO | 17.87964038 | 12.9572827 | −0.464554521 | 2.75E−06 | 4.46E−06 |
| POLR2J | 16.07014268 | 32.76108605 | 1.027600446 | 2.75E−19 | 1.45E−18 |
| FHIT | 0.791113118 | 2.26275842 | 1.516126666 | 6.30E−22 | 5.60E−21 |
| GPD1 | 36.7540008 | 17.69852929 | −1.05427182 | 4.63E−13 | 1.29E−12 |
| CHST12 | 0.378759634 | 0.713920394 | 0.91448063 | 2.32E−13 | 6.63E−13 |
| HMGCS1 | 26.26538034 | 41.55101209 | 0.661721142 | 0.000113855 | 0.000159481 |
| BHMT | 188.5596978 | 89.71365222 | −1.071621902 | 2.19E−13 | 6.30E−13 |
| UGT1A4 | 30.3252135 | 24.22035055 | −0.324298057 | 6.85E−08 | 1.21E−07 |
| LAP3 | 38.7008652 | 30.80759113 | −0.329079938 | 2.01E−05 | 3.02E−05 |
| ACP4 | 0.013881318 | 0.395615568 | 4.83288278 | 1.58E−12 | 4.10E−12 |
| PDE11A | 0.6499311 | 0.489365997 | −0.409372922 | 4.45E−07 | 7.54E−07 |
| POLA1 | 0.530661838 | 1.544546194 | 1.541318312 | 4.30E−19 | 2.19E−18 |
| ALDH2 | 154.9258366 | 71.34767851 | −1.118639365 | 1.47E−22 | 1.42E−21 |
| CES5A | 1.156581576 | 0.643541115 | −0.845762796 | 2.67E−09 | 5.18E−09 |
| XDH | 15.64604762 | 9.530809199 | −0.715127647 | 1.12E−09 | 2.25E−09 |
| JMJD7-PLA2G4B | 0.08957713 | 0.24466282 | 1.449592525 | 1.59E−10 | 3.50E−10 |
| HK1 | 1.756529668 | 3.532845207 | 1.008102594 | 0.004998106 | 0.006106101 |
| POLR3A | 1.148311722 | 2.372790232 | 1.047070233 | 7.34E−23 | 7.69E−22 |
| ACSM3 | 8.26916818 | 3.080383675 | −1.424632156 | 3.51E−21 | 2.50E−20 |
| ZNRD1 | 1.920201268 | 4.468929856 | 1.218671864 | 4.96E−20 | 2.88E−19 |
| GFPT1 | 4.77169852 | 8.360365723 | 0.809063159 | 2.23E−11 | 5.31E−11 |
| LCLAT1 | 1.079431836 | 1.825327458 | 0.757883159 | 4.13E−11 | 9.55E−11 |
| SULT1E1 | 4.766748444 | 3.612816882 | −0.399881359 | 7.85E−08 | 1.38E−07 |
| LIPG | 4.0745132 | 2.953168638 | −0.464363962 | 7.83E−06 | 1.22E−05 |
| MBOAT7 | 4.30759108 | 10.22851875 | 1.247644029 | 2.35E−24 | 3.77E−23 |
| PCYT2 | 15.79421824 | 27.65662255 | 0.808228454 | 6.16E−10 | 1.27E−09 |
| GMPS | 2.12571058 | 5.125880879 | 1.269854768 | 3.78E−23 | 4.40E−22 |
| SGPL1 | 5.76193592 | 8.91802992 | 0.630171424 | 5.88E−09 | 1.11E−08 |
| PYCR1 | 0.861568492 | 8.121217599 | 3.236658648 | 9.35E−06 | 1.45E−05 |
| PIPOX | 96.0303444 | 72.54766923 | −0.404561087 | 4.98E−06 | 7.84E−06 |
| MDH1 | 20.8540302 | 33.17323875 | 0.669693649 | 3.56E−14 | 1.07E−13 |
| PAFAH1B3 | 2.183462234 | 13.95513656 | 2.676106759 | 2.23E−19 | 1.19E−18 |
| ETNK1 | 3.83300098 | 5.261446108 | 0.456985012 | 0.000160517 | 0.000221554 |
| PRPS2 | 6.2703327 | 8.859932778 | 0.498753759 | 0.003710583 | 0.004615306 |
| ENTPD8 | 4.7108131 | 3.744468425 | −0.33121517 | 1.12E−05 | 1.73E−05 |
| ALOX12 | 0.109011063 | 0.275266308 | 1.336353479 | 1.04E−12 | 2.76E−12 |
| ACACB | 8.46489348 | 5.654933634 | −0.581981821 | 3.48E−09 | 6.71E−09 |
| GMDS | 3.66597734 | 7.7145025 | 1.073375254 | 3.91E−10 | 8.20E−10 |
| ME3 | 0.531760212 | 1.606608404 | 1.595170589 | 3.70E−15 | 1.21E−14 |

| Gene | conMean | treatMean | logFC | pValue | fdr |
|---|---|---|---|---|---|
| PIK3CA | 1.177370638 | 1.455137701 | 0.305587128 | 0.039746951 | 0.045262365 |
| DCK | 1.436440984 | 3.648930365 | 1.344974898 | 1.32E−15 | 4.39E−15 |
| GYS1 | 1.954834784 | 4.327244832 | 1.146402068 | 3.57E−18 | 1.63E−17 |
| ADO | 2.54632174 | 3.430583608 | 0.430039305 | 5.05E−05 | 7.37E−05 |
| AGPAT1 | 8.15078736 | 20.35100554 | 1.320088745 | 2.02E−24 | 3.31E−23 |
| ADSL | 3.82300396 | 8.444661914 | 1.143332971 | 6.03E−24 | 8.28E−23 |
| ALDH6A1 | 64.384127 | 28.49727636 | −1.175881016 | 9.37E−20 | 5.28E−19 |
| INPP5A | 3.35817814 | 4.788868444 | 0.512006041 | 1.30E−07 | 2.25E−07 |
| MTMR7 | 0.146421709 | 0.848734424 | 2.535183728 | 1.88E−05 | 2.83E−05 |
| SULT1A1 | 13.47559808 | 9.050285627 | −0.574314075 | 3.86E−09 | 7.41E−09 |
| CYP26A1 | 6.569500214 | 0.658808858 | −3.31785176 | 1.48E−21 | 1.21E−20 |
| PDHB | 10.74619842 | 14.41091844 | 0.423335903 | 4.93E−09 | 9.42E−09 |
| ALDH1B1 | 67.6148248 | 40.63371028 | −0.734662494 | 5.75E−11 | 1.32E−10 |
| ACOX1 | 33.2083646 | 25.9762862 | −0.354351492 | 1.12E−07 | 1.94E−07 |
| PCYT1A | 3.16719152 | 5.454631679 | 0.784277673 | 2.39E−20 | 1.42E−19 |
| NEU1 | 7.53752836 | 26.66679468 | 1.822880995 | 3.40E−28 | 3.93E−26 |
| POLR3G | 0.27411095 | 0.623002221 | 1.184477344 | 3.97E−13 | 1.11E−12 |
| GPD1L | 0.656858692 | 1.902400678 | 1.534166188 | 5.88E−09 | 1.11E−08 |
| TSTA3 | 11.35656626 | 26.20332018 | 1.206222933 | 9.59E−18 | 4.11E−17 |
| AK1 | 2.37631508 | 3.937539076 | 0.728568102 | 1.41E−08 | 2.57E−08 |
| PRODH2 | 39.0309294 | 28.08474289 | −0.474831221 | 1.12E−07 | 1.94E−07 |
| AOC1 | 1.798183299 | 0.745771443 | −1.269734631 | 0.001027935 | 0.001333489 |
| SPTLC2 | 1.75140612 | 2.742051601 | 0.646742062 | 3.29E−10 | 6.95E−10 |
| KHK | 82.384005 | 61.54386532 | −0.420749206 | 1.71E−07 | 2.97E−07 |
| DGKQ | 1.457440362 | 4.03086916 | 1.467654105 | 4.12E−23 | 4.65E−22 |
| ACAA2 | 105.504459 | 52.75512735 | −0.99992075 | 5.19E−20 | 2.99E−19 |
| ADPRM | 2.7431114 | 3.986350382 | 0.53925531 | 1.35E−05 | 2.06E−05 |
| AGXT2 | 21.68184318 | 9.089264569 | −1.254251933 | 2.47E−18 | 1.14E−17 |
| TYMP | 15.65976866 | 31.83914656 | 1.023738765 | 1.52E−06 | 2.49E−06 |
| SYNJ2 | 1.00822186 | 2.334073703 | 1.211036978 | 8.08E−10 | 1.65E−09 |
| UPB1 | 32.3556494 | 25.21832405 | −0.359545233 | 0.00015816 | 0.000219102 |
| ENTPD2 | 0.372271672 | 1.96584486 | 2.400721728 | 3.97E−12 | 9.92E−12 |
| GPX1 | 124.116905 | 222.3074623 | 0.840856749 | 1.77E−10 | 3.88E−10 |
| PRPS1 | 12.66083042 | 18.44318764 | 0.542715993 | 3.69E−06 | 5.87E−06 |
| GLB1 | 9.81417212 | 18.17416942 | 0.888950954 | 8.00E−18 | 3.51E−17 |
| DHDH | 0.083578928 | 0.534516557 | 2.677023475 | 3.88E−10 | 8.16E−10 |
| PTGES2 | 7.21657504 | 16.7167287 | 1.211906346 | 8.10E−22 | 7.03E−21 |
| NME1 | 5.12519878 | 17.66287993 | 1.785040727 | 4.09E−24 | 5.94E−23 |
| GLYCTK | 55.138394 | 52.81353627 | −0.062149503 | 0.028147703 | 0.032445062 |
| TKFC | 25.9085641 | 18.57767451 | −0.479859139 | 9.44E−08 | 1.65E−07 |
| ADCY3 | 0.561033466 | 1.082943581 | 0.948799348 | 0.000235869 | 0.000319141 |
| HAGHL | 0.048049798 | 0.557557535 | 3.536518411 | 1.02E−21 | 8.52E−21 |

| Gene | conMean | treatMean | logFC | *p*Value | fdr |
|------|---------|-----------|-------|----------|-----|
| UGT2A1 | 0.98181545 | 0.696545876 | −0.495233494 | 0.000447104 | 0.000590145 |
| PTGS2 | 0.685695488 | 0.209820152 | −1.708414783 | 5.31E−17 | 2.01E−16 |
| ALDOB | 2772.70035 | 1401.228338 | −0.98459964 | 1.94E−14 | 5.97E−14 |
| TYRP1 | 0.008600457 | 0.267724997 | 4.960194749 | 0.000375099 | 0.000501238 |
| NUDT9 | 8.49499344 | 10.95659974 | 0.367115405 | 0.000437831 | 0.000578918 |
| MPI | 2.65308182 | 4.29403827 | 0.694665881 | 6.65E−13 | 1.80E−12 |
| PPAT | 0.863819914 | 1.765653998 | 1.031400175 | 3.59E−13 | 1.02E−12 |
| GPD2 | 0.764990188 | 1.774788278 | 1.214133781 | 1.94E−12 | 4.95E−12 |
| DTYMK | 2.81252568 | 11.25842326 | 2.001066619 | 4.78E−28 | 4.51E−26 |
| GBA3 | 29.65604208 | 9.090840342 | −1.705840501 | 2.15E−20 | 1.29E−19 |
| PGS1 | 1.087267554 | 2.837714227 | 1.384022308 | 3.10E−24 | 4.88E−23 |
| ITPA | 8.35692768 | 20.76255349 | 1.312939329 | 2.62E−22 | 2.42E−21 |
| GSTA2 | 171.5788165 | 122.8201096 | −0.482324649 | 3.52E−06 | 5.61E−06 |
| ENPP2 | 4.152915006 | 9.060232331 | 1.1254237 | 9.16E−05 | 0.000130485 |
| GLDC | 19.3329667 | 15.99389614 | −0.273541615 | 0.001078603 | 0.001394427 |
| PKM | 4.01863632 | 25.24352982 | 2.65113573 | 1.86E−14 | 5.79E−14 |
| MTMR6 | 2.721699042 | 3.270919686 | 0.26518879 | 0.033367063 | 0.038285916 |
| UPP1 | 2.553424264 | 3.724478367 | 0.544605114 | 0.000545856 | 0.000715488 |
| RRM2 | 0.469398657 | 6.378298565 | 3.764286013 | 3.18E−26 | 1.33E−24 |
| ACSL6 | 0.2617771 | 1.020156731 | 1.962380012 | 0.035460273 | 0.040503035 |
| LDHA | 83.2189104 | 80.19834618 | −0.053338913 | 0.042299176 | 0.047951769 |
| GPI | 20.7880496 | 42.19970111 | 1.021478374 | 1.93E−16 | 7.02E−16 |
| ACSM5 | 50.4663964 | 20.22345997 | −1.319293229 | 2.44E−17 | 9.70E−17 |
| PMM1 | 7.81445296 | 10.63144872 | 0.444121415 | 0.001111858 | 0.001432513 |
| HADH | 41.8828638 | 35.57719138 | −0.235407469 | 5.47E−05 | 7.95E−05 |
| SAT2 | 85.0755162 | 76.44819578 | −0.154261546 | 0.006385788 | 0.00776372 |
| GNE | 25.03165064 | 13.02163252 | −0.942843099 | 2.36E−17 | 9.45E−17 |
| GSTM2 | 0.271784875 | 0.548345501 | 1.012620017 | 0.000818116 | 0.001066801 |
| AKR1C2 | 16.24967136 | 50.10020952 | 1.624406096 | 5.01E−06 | 7.87E−06 |
| CDO1 | 96.0541018 | 74.41269956 | −0.368298364 | 2.75E−05 | 4.08E−05 |
| CNDP1 | 6.560749296 | 1.024432657 | −2.679035444 | 1.45E−22 | 1.42E−21 |
| AHCYL1 | 15.2581398 | 20.6807703 | 0.438710837 | 2.04E−05 | 3.05E−05 |
| ACP6 | 0.71604374 | 1.761317951 | 1.298535743 | 2.41E−20 | 1.42E−19 |
| NME7 | 0.643526996 | 1.362272741 | 1.081942999 | 4.98E−17 | 1.90E−16 |
| ADH1C | 476.144326 | 233.435638 | −1.02837411 | 1.14E−12 | 3.00E−12 |
| TBXAS1 | 1.383732803 | 1.149415109 | −0.267665469 | 0.000395307 | 0.000526379 |
| BAAT | 190.9706308 | 179.3099227 | −0.090895458 | 0.01578084 | 0.018472146 |
| AOC2 | 0.219990869 | 0.495573818 | 1.171656326 | 1.73E−09 | 3.44E−09 |
| AGL | 6.99793654 | 3.850530943 | −0.861872192 | 9.11E−17 | 3.36E−16 |
| IDO2 | 0.862516082 | 0.34741407 | −1.311895173 | 9.61E−15 | 3.06E−14 |
| UMPS | 3.1273573 | 4.722024104 | 0.59446135 | 3.46E−13 | 9.83E−13 |
| RDH11 | 16.88686872 | 20.31333093 | 0.26652499 | 0.041311007 | 0.04690197 |

**Table 1** (*continued*)

| Gene | conMean | treatMean | logFC | *p*Value | fdr |
|------|---------|-----------|-------|----------|-----|
| AADAT | 11.0372536 | 2.431650612 | −2.182373373 | 7.44E−26 | 2.16E−24 |
| ME1 | 2.644506866 | 8.410134567 | 1.669130164 | 0.000433795 | 0.000574588 |
| ACSM2A | 35.2888359 | 22.45481559 | −0.652186966 | 3.27E−08 | 5.87E−08 |
| CYP2C8 | 449.461848 | 100.5747926 | −2.159929897 | 6.04E−26 | 1.82E−24 |
| LCMT1 | 2.3080183 | 5.778087404 | 1.323937364 | 5.58E−25 | 1.20E−23 |
| PLA2G12A | 5.69083708 | 5.41609293 | −0.071388385 | 0.004386592 | 0.005385166 |
| AZIN2 | 0.124387656 | 0.277501557 | 1.157652541 | 9.96E−10 | 2.02E−09 |
| PTGIS | 2.191950054 | 1.166763256 | −0.909703067 | 2.41E−14 | 7.37E−14 |
| GPX4 | 118.53575 | 188.6938862 | 0.670725443 | 1.28E−10 | 2.86E−10 |
| RRM1 | 3.92014216 | 10.15498263 | 1.373209893 | 1.84E−21 | 1.42E−20 |
| ACAA1 | 48.7472452 | 25.20723697 | −0.951482702 | 4.88E−18 | 2.21E−17 |
| NNMT | 469.3829578 | 140.666585 | −1.738485801 | 1.15E−17 | 4.88E−17 |
| TDO2 | 85.2300468 | 35.60804767 | −1.259158787 | 2.05E−15 | 6.71E−15 |
| ADCY6 | 1.052476086 | 3.845742664 | 1.869474775 | 3.47E−25 | 7.71E−24 |
| SARDH | 34.544682 | 18.5565624 | −0.896534155 | 1.76E−17 | 7.30E−17 |
| RFK | 5.10336714 | 7.08934908 | 0.474203736 | 1.99E−05 | 3.00E−05 |
| GAMT | 189.2736142 | 172.100408 | −0.137222788 | 0.0195276 | 0.022717008 |
| ACYP1 | 0.435832892 | 1.541828841 | 1.822795635 | 4.19E−26 | 1.44E−24 |
| MTHFD1 | 36.8332948 | 23.39997829 | −0.654503263 | 3.30E−11 | 7.65E−11 |
| GNMT | 134.7991696 | 71.11434267 | −0.922599145 | 2.73E−11 | 6.39E−11 |
| POLR2E | 23.3947028 | 33.18577334 | 0.504382993 | 1.39E−11 | 3.40E−11 |
| PAFAH2 | 5.46319746 | 4.725647625 | −0.209233512 | 7.56E−05 | 0.000108525 |
| RRM2B | 3.53135628 | 5.709601422 | 0.693167654 | 1.77E−08 | 3.21E−08 |
| CYP1A2 | 162.0610482 | 24.79812246 | −2.708234578 | 4.83E−24 | 6.89E−23 |
| ALDH18A1 | 7.39242978 | 15.6829565 | 1.085077018 | 4.28E−15 | 1.39E−14 |
| HDC | 0.257674991 | 0.232402497 | −0.148926952 | 0.000172793 | 0.000238063 |
| GANC | 0.61022308 | 1.019814907 | 0.740898679 | 3.01E−10 | 6.41E−10 |
| ADCY1 | 1.645927202 | 1.296404921 | −0.344384126 | 1.17E−09 | 2.34E−09 |
| FBP1 | 394.352886 | 138.9132655 | −1.505302825 | 4.76E−21 | 3.27E−20 |
| POLR1C | 3.35005178 | 6.978792797 | 1.058794103 | 2.33E−21 | 1.74E−20 |
| GLS | 1.201759816 | 4.164693323 | 1.793061676 | 3.87E−12 | 9.70E−12 |
| LPCAT2 | 0.474627134 | 1.255891704 | 1.40384558 | 0.000234735 | 0.000318178 |
| P4HA2 | 1.008157448 | 4.303512033 | 2.093793535 | 1.64E−22 | 1.57E−21 |
| CKB | 3.60284462 | 23.67772586 | 2.716322184 | 3.20E−06 | 5.13E−06 |
| ITPKA | 0.460224664 | 4.393506423 | 3.254962596 | 5.55E−18 | 2.48E−17 |
| POLR2C | 9.07856552 | 14.29939552 | 0.655417896 | 2.19E−10 | 4.77E−10 |
| COX10 | 2.5153264 | 3.46926254 | 0.4638834 | 0.000317057 | 0.000425939 |
| POLR2L | 45.7742558 | 93.97066345 | 1.037674004 | 3.18E−19 | 1.66E−18 |
| ALDH3B1 | 1.04183392 | 2.518889866 | 1.27366273 | 1.40E−06 | 2.30E−06 |
| ITPKB | 0.905496968 | 1.49022587 | 0.718749297 | 4.85E−05 | 7.09E−05 |
| TXNRD1 | 6.76314148 | 28.10662419 | 2.055144747 | 1.93E−17 | 7.90E−17 |
| CA5B | 0.142795734 | 0.550149828 | 1.945871693 | 6.48E−16 | 2.22E−15 |

**Table 1** (*continued*)

| Gene | conMean | treatMean | logFC | pValue | fdr |
| --- | --- | --- | --- | --- | --- |
| PLCB1 | 0.225675628 | 1.091508615 | 2.274000988 | 4.77E−14 | 1.42E−13 |
| NUDT5 | 9.44814408 | 17.61621736 | 0.898801304 | 1.61E−21 | 1.30E−20 |
| ODC1 | 18.19301946 | 33.29143499 | 0.871766053 | 1.65E−08 | 3.00E−08 |
| ACACA | 1.393724136 | 3.919558477 | 1.491746117 | 6.37E−20 | 3.62E−19 |
| INPP5J | 0.03995144 | 0.299135465 | 2.904479538 | 2.70E−11 | 6.36E−11 |
| DNMT3B | 0.128279551 | 0.715787579 | 2.480240297 | 7.59E−21 | 4.97E−20 |
| PDE7A | 0.574264186 | 1.45451893 | 1.340755577 | 3.24E−14 | 9.82E−14 |
| INPPL1 | 4.00731938 | 9.917278334 | 1.30730675 | 4.82E−21 | 3.28E−20 |
| POLR3F | 1.290860464 | 2.890671173 | 1.163071445 | 2.94E−25 | 6.72E−24 |
| DPYS | 89.3997666 | 56.69502639 | −0.657048886 | 2.46E−09 | 4.82E−09 |
| CBR3 | 0.238376351 | 1.432379474 | 2.587100731 | 8.94E−16 | 3.01E−15 |
| CA2 | 43.516094 | 27.05355184 | −0.68573105 | 1.63E−13 | 4.71E−13 |
| PLA2G1B | 0.358601163 | 2.766570668 | 2.947646705 | 2.58E−09 | 5.01E−09 |
| NIT2 | 8.36875624 | 7.98024058 | −0.068580986 | 0.03755159 | 0.042826964 |
| PAFAH1B2 | 5.55860366 | 8.178248219 | 0.557069332 | 2.81E−10 | 6.00E−10 |
| GYS2 | 38.5893622 | 12.55410139 | −1.620044433 | 4.34E−21 | 3.00E−20 |
| GLUL | 58.1799336 | 366.9451525 | 2.656970884 | 0.000190569 | 0.000261125 |
| GPT2 | 34.4213786 | 29.04180666 | −0.245173675 | 0.000173655 | 0.000238815 |
| FMO2 | 0.513662603 | 0.392428653 | −0.388390661 | 1.83E−05 | 2.78E−05 |
| TXNDC12 | 10.49130104 | 17.80804426 | 0.763335484 | 7.64E−21 | 4.97E−20 |
| MTMR2 | 0.825414542 | 1.902368228 | 1.204605765 | 2.99E−11 | 6.94E−11 |
| AHCYL2 | 1.248491948 | 1.918117883 | 0.619504871 | 8.41E−07 | 1.40E−06 |
| ALDOA | 20.55360858 | 72.12748631 | 1.811157437 | 2.08E−19 | 1.12E−18 |
| MDH2 | 46.934258 | 79.50191043 | 0.760348177 | 6.89E−16 | 2.34E−15 |
| FADS2 | 9.642111654 | 20.05759983 | 1.056727937 | 0.011186336 | 0.013321267 |
| PAPSS1 | 2.28088236 | 5.466021302 | 1.260899043 | 7.56E−17 | 2.81E−16 |
| NANP | 0.94921358 | 1.727478428 | 0.863863049 | 2.01E−15 | 6.61E−15 |
| MAT1A | 492.371978 | 208.7499778 | −1.237972613 | 2.22E−21 | 1.68E−20 |
| GSTA4 | 3.88265552 | 12.89632448 | 1.73184433 | 2.91E−16 | 1.03E−15 |
| ADCY4 | 0.314709281 | 0.7262269 | 1.206400646 | 6.53E−13 | 1.77E−12 |
| INPP4B | 0.149343996 | 0.294366749 | 0.97897548 | 0.001456753 | 0.001857852 |
| UGT1A10 | 0.008250359 | 0.550736449 | 6.060761396 | 0.01163501 | 0.013812001 |
| NOS2 | 0.097262382 | 0.402505335 | 2.049054083 | 5.21E−13 | 1.44E−12 |
| ACER3 | 0.859076874 | 1.427533057 | 0.732665013 | 9.70E−08 | 1.69E−07 |
| PIK3C3 | 0.909520322 | 1.266871369 | 0.47809227 | 2.98E−07 | 5.07E−07 |
| HAAO | 80.8500472 | 45.99512303 | −0.813767719 | 3.64E−12 | 9.15E−12 |
| DEGS1 | 7.60895486 | 17.27093585 | 1.182576051 | 6.59E−18 | 2.93E−17 |
| NUDT2 | 7.28607756 | 17.12103503 | 1.232555663 | 4.35E−19 | 2.20E−18 |
| ADH1B | 527.994414 | 221.1341117 | −1.255601077 | 3.38E−17 | 1.31E−16 |
| POLR2I | 11.61097278 | 19.34012161 | 0.736108019 | 2.47E−10 | 5.33E−10 |
| NT5M | 0.40577981 | 1.573026546 | 1.954774029 | 3.24E−17 | 1.27E−16 |
| GATM | 201.0644772 | 134.4996308 | −0.580056005 | 1.14E−08 | 2.11E−08 |
| PLPP3 | 35.3993676 | 17.43031475 | −1.022124966 | 3.08E−17 | 1.22E−16 |

**Table 1** (*continued*)

| Gene | conMean | treatMean | logFC | *p*Value | fdr |
|---|---|---|---|---|---|
| PHPT1 | 13.54757584 | 48.38776828 | 1.836607678 | 3.87E−26 | 1.43E−24 |
| AKR1B10 | 21.56249119 | 348.4054764 | 4.014171528 | 1.73E−11 | 4.18E−11 |
| DGKA | 0.329520905 | 0.633643682 | 0.943301801 | 2.37E−05 | 3.54E−05 |
| TREH | 1.851689492 | 1.32430031 | −0.483611878 | 3.78E−06 | 6.00E−06 |
| UGT1A6 | 2.955261072 | 7.241389897 | 1.292981046 | 0.006980058 | 0.008405017 |
| B4GALT2 | 5.90955282 | 11.4905014 | 0.959320883 | 1.47E−17 | 6.13E−17 |
| SGMS2 | 3.765996564 | 1.802598975 | −1.062953208 | 1.68E−05 | 2.55E−05 |
| ENTPD1 | 0.634465836 | 1.48168498 | 1.223624363 | 2.79E−19 | 1.46E−18 |
| INPP5E | 1.353489428 | 3.225346035 | 1.252770331 | 1.93E−19 | 1.05E−18 |
| AMPD2 | 4.28833292 | 7.542024427 | 0.814534911 | 7.54E−13 | 2.02E−12 |
| KYNU | 2.4308026 | 2.247638585 | −0.113022668 | 9.84E−07 | 1.63E−06 |
| UGP2 | 58.5313736 | 40.07029916 | −0.546676855 | 1.31E−10 | 2.91E−10 |
| CDA | 29.0720696 | 17.30167458 | −0.7487221 | 1.20E−10 | 2.69E−10 |
| NME1-NME2 | 0.318942837 | 0.531576005 | 0.736978107 | 0.000431791 | 0.000572938 |
| DGUOK | 11.27155494 | 19.37659762 | 0.781628714 | 1.50E−13 | 4.35E−13 |
| AK2 | 14.9739754 | 17.94421295 | 0.261061357 | 0.000889768 | 0.001158232 |
| PI4KA | 2.861807906 | 5.047843187 | 0.818740256 | 2.57E−10 | 5.53E−10 |
| COX15 | 6.69914978 | 8.729761864 | 0.381964291 | 2.57E−08 | 4.65E−08 |
| CHST13 | 13.29861354 | 25.15093577 | 0.919336234 | 5.40E−07 | 9.06E−07 |
| GLO1 | 32.9487238 | 51.30883037 | 0.638984552 | 5.19E−09 | 9.87E−09 |
| HAO2 | 86.8433924 | 25.34855366 | −1.776512651 | 1.58E−20 | 9.67E−20 |
| GCDH | 24.69117048 | 14.79007361 | −0.739365996 | 2.95E−12 | 7.47E−12 |
| CPT1B | 0.121235197 | 0.398590265 | 1.717097872 | 1.74E−14 | 5.44E−14 |
| SPTLC1 | 3.060500882 | 4.809781218 | 0.652203488 | 1.51E−05 | 2.29E−05 |
| PYGB | 4.35291446 | 18.82163409 | 2.112338313 | 8.70E−26 | 2.35E−24 |
| POLR2H | 5.75387058 | 12.47900765 | 1.116898537 | 1.04E−22 | 1.07E−21 |
| POLR2B | 4.42658506 | 7.625983009 | 0.784729173 | 7.24E−11 | 1.65E−10 |
| RETSAT | 38.7021028 | 37.57376368 | −0.04268632 | 0.019656262 | 0.022831504 |
| PTDSS2 | 1.662328 | 4.164425941 | 1.324912563 | 2.15E−23 | 2.57E−22 |
| PYCR2 | 5.37894786 | 15.26388831 | 1.50472661 | 5.48E−28 | 4.59E−26 |
| METTL2B | 1.507055692 | 2.85584579 | 0.922185347 | 1.48E−21 | 1.21E−20 |
| DGKH | 0.113162972 | 0.327426191 | 1.532767749 | 3.75E−16 | 1.31E−15 |
| GSTA1 | 630.2036156 | 548.263622 | −0.200948279 | 0.00110705 | 0.001428757 |
| MAT2B | 7.7057353 | 10.40578761 | 0.433381633 | 4.64E−06 | 7.33E−06 |
| BPNT1 | 5.53523458 | 10.96770642 | 0.986545495 | 3.39E−21 | 2.44E−20 |
| DNMT3A | 0.555289364 | 2.215351506 | 1.996223959 | 7.94E−24 | 1.07E−22 |
| SEPHS2 | 127.9557976 | 190.1067404 | 0.571164168 | 2.59E−10 | 5.56E−10 |
| POLE3 | 8.66220048 | 16.03828744 | 0.888714631 | 6.27E−19 | 3.13E−18 |
| INPP4A | 0.504742884 | 0.934387362 | 0.888472096 | 7.54E−11 | 1.70E−10 |
| CYP2E1 | 530.2645806 | 266.8712181 | −0.990568666 | 5.25E−11 | 1.21E−10 |
| CYP2J2 | 34.1905172 | 23.20978966 | −0.558862799 | 5.17E−11 | 1.19E−10 |
| ITPK1 | 7.76110164 | 15.32881022 | 0.981912369 | 6.29E−16 | 2.16E−15 |
| POLR2D | 3.37104148 | 5.942900711 | 0.817972898 | 1.73E−20 | 1.05E−19 |

**Table 1** (*continued*)

| Gene | conMean | treatMean | logFC | *p*Value | fdr |
|------|---------|-----------|-------|----------|-----|
| CSAD | 8.06462968 | 6.521514989 | −0.306401137 | 7.35E−05 | 0.000105697 |
| NADSYN1 | 1.32031081 | 2.690367867 | 1.026925864 | 1.07E−21 | 8.84E−21 |
| AMD1 | 4.19170934 | 5.335768195 | 0.348157311 | 0.011707199 | 0.01387588 |
| GSTM5 | 0.363642937 | 0.231850861 | −0.649325476 | 3.60E−10 | 7.60E−10 |
| GAL3ST1 | 0.673563029 | 6.610136193 | 3.294795138 | 0.00012214 | 0.000170769 |
| GPT | 47.2407036 | 38.45088064 | −0.297013812 | 4.98E−06 | 7.84E−06 |
| AGPAT3 | 8.2516119 | 11.64639678 | 0.497135802 | 4.54E−07 | 7.67E−07 |
| POLR3B | 2.42816858 | 2.951770627 | 0.281712031 | 0.003889012 | 0.004829283 |
| ASL | 53.904885 | 52.08163964 | −0.049641151 | 0.001055415 | 0.001366789 |
| PLA2G15 | 2.72944324 | 4.31619806 | 0.661154372 | 5.77E−08 | 1.02E−07 |
| BDH2 | 10.50709572 | 6.470275333 | −0.699464937 | 1.71E−12 | 4.41E−12 |
| DGKD | 0.77171494 | 1.905129995 | 1.303749501 | 4.07E−19 | 2.09E−18 |
| ADI1 | 91.0607558 | 69.40997785 | −0.391686365 | 2.37E−07 | 4.07E−07 |
| CHPT1 | 14.3208144 | 19.59827237 | 0.452612945 | 1.36E−06 | 2.24E−06 |
| NME3 | 9.19139972 | 22.42554503 | 1.286786564 | 6.38E−22 | 5.60E−21 |
| DHODH | 15.80542924 | 6.798435324 | −1.217145566 | 4.99E−16 | 1.73E−15 |
| LCAT | 68.1154072 | 16.82335661 | −2.01751558 | 2.17E−26 | 9.64E−25 |
| GLS2 | 4.778706618 | 1.78680446 | −1.419238437 | 1.24E−15 | 4.17E−15 |
| PLPP1 | 14.99118618 | 30.87743991 | 1.042438601 | 9.08E−10 | 1.85E−09 |
| CKMT1B | 0.007144548 | 0.294667545 | 5.366101652 | 0.004188776 | 0.005159096 |
| ENTPD4 | 1.775126334 | 2.540169076 | 0.517002824 | 0.002229478 | 0.002814809 |
| PLCD1 | 1.691310116 | 2.545895876 | 0.590032202 | 2.56E−07 | 4.37E−07 |
| G6PC | 204.0482294 | 152.8745478 | −0.416561961 | 7.14E−06 | 1.12E−05 |
| ATIC | 7.75802162 | 18.69461756 | 1.268862257 | 1.56E−23 | 1.96E−22 |
| PEMT | 52.8566338 | 30.87396532 | −0.775693761 | 4.96E−14 | 1.47E−13 |
| AMY2B | 0.407575554 | 0.944786186 | 1.212920347 | 3.39E−06 | 5.43E−06 |
| MGST1 | 91.208676 | 75.16583894 | −0.279093923 | 0.000239299 | 0.000323204 |
| CBS | 1.188079795 | 0.956133191 | −0.313348227 | 0.028413017 | 0.032700957 |
| PFKFB4 | 0.151297184 | 0.756509371 | 2.321972815 | 5.38E−18 | 2.42E−17 |
| MTHFD1L | 0.557753522 | 2.524923016 | 2.178539778 | 2.35E−22 | 2.19E−21 |
| SUOX | 10.01822034 | 9.128650786 | −0.134152697 | 0.011544688 | 0.013726361 |
| ACP2 | 20.4062632 | 24.48232224 | 0.262728389 | 0.004059381 | 0.005016093 |
| GOT1 | 154.184527 | 113.1036936 | −0.447011949 | 1.05E−06 | 1.73E−06 |
| MCEE | 8.66946072 | 8.069239048 | −0.103509624 | 0.013633303 | 0.016032934 |
| POLR3C | 2.99792546 | 7.304838 | 1.284887766 | 1.45E−22 | 1.42E−21 |

+ (-0.1116 * ACADS expression) + (0.1978 * UCK2 expression) + (-0.0143 * GOT2 expression) + (-0.0295 * ADH4 expression) + (-0.3244 * LDHA expression) + (0.0520 * ME1 expression) + (0.0105 * TXNRD1 expression) + (0.0433 * B4GALT2 expression) + (0.1975* AK2 expression) + (0.1783* PTDSS2 expression) + (-0.023 * CSAD expression) + (0.0207 * AMD1 expression). Based on the median of the risk score, the sample was rated as low and high risk (Fig. 4A). Of these, 164 patients were in the high-risk group and 165 patients were in the low-risk group. Principal component analysis (PCA) and t-distribution random neighborhood embedding (t-SNE) analysis showed that patients in

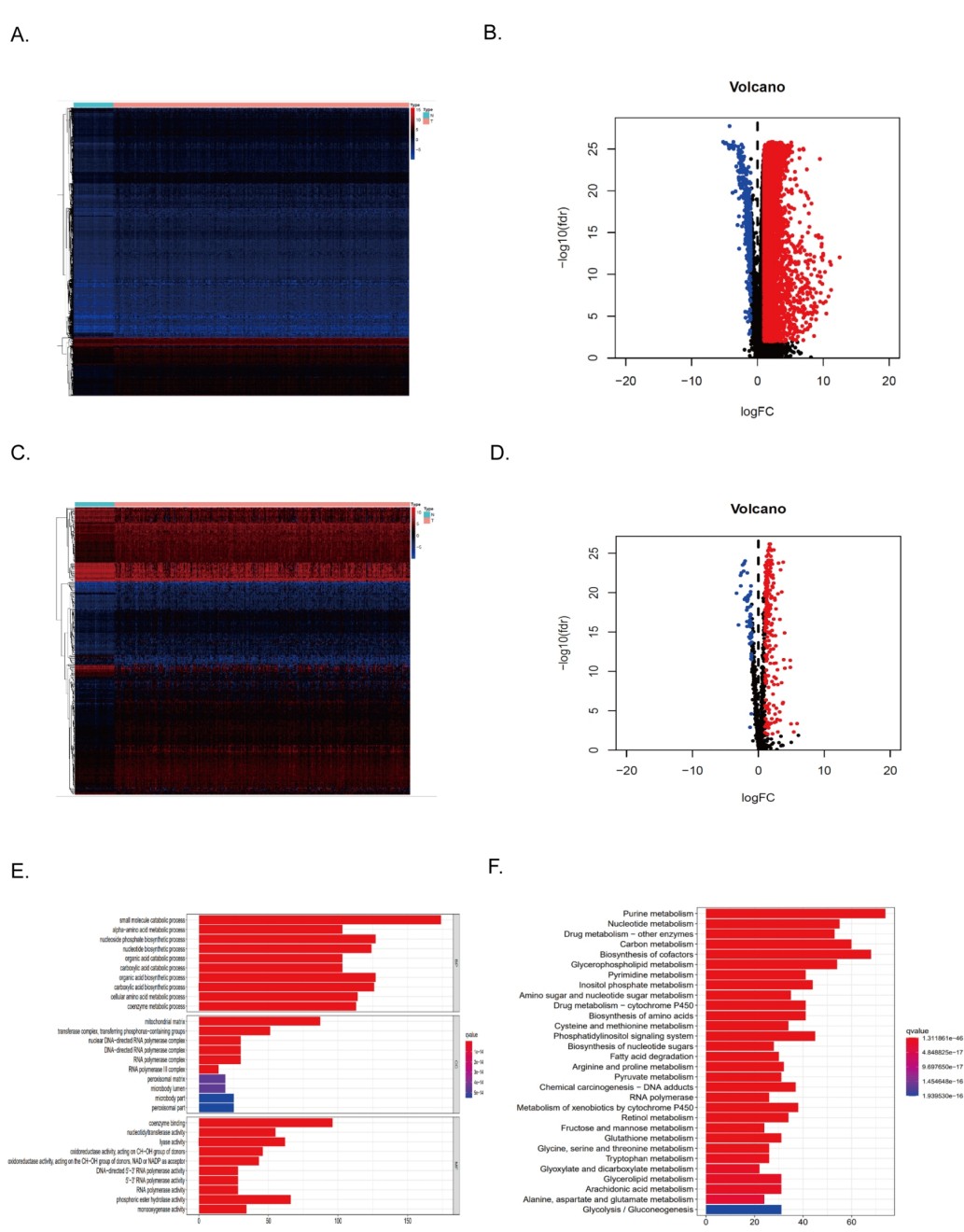

**Figure 2** **Differentially expressed MRGs in HCC patients.** (A) Heatmap of DEGs in HCC. The color from blue to red represents the progression from low expression to high expression. (B) Volcano plot of differentially expressed genes in HCC. The red dots in the plot represent upregulated genes, and the blue dots represent downregulated genes with statistical significance. Black dots represent no differentially expressed genes in HCC. (C) Heatmap of differentially expressed MRGs in HCC. The color from blue to red represents the progression from low expression to high expression. (D) Volcano plot of differentially expressed MRGs in HCC.

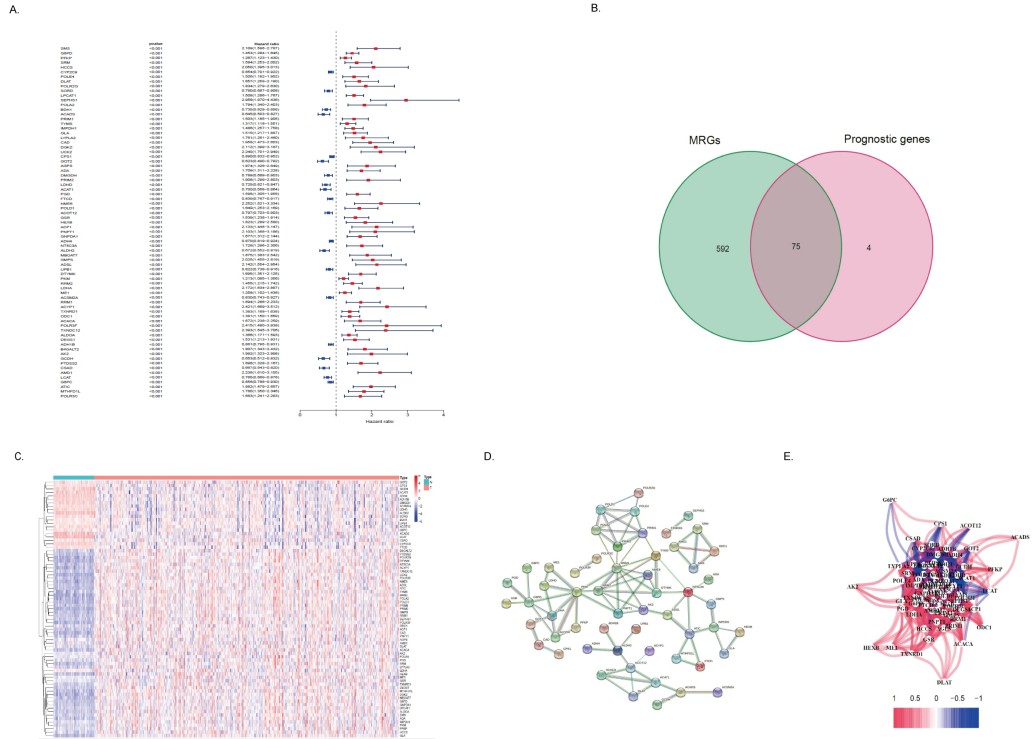

**Figure 3 Identification of the metabolism-related DEGs with prognostic values in the TCGA cohort.**
(A) Forest plot shows the results of univariate Cox regression analysis between gene expression and OS.
(B) Venn diagram shows the intersection of differentially expressed MRGs and prognostic-related genes.
(B) The heatmap shows the expression difference of these 75 MRGs with a prognostic value. (D) The PPI
network showed the interaction between 75 genes. (E) A correlation network of 75 MRGs with prognostic
value. The correlation coefficient is distinguished by different colors.

different risk groups were distributed in two directions (Figs. 4B–4C). This suggests that
the MRRS model can distinguish well between high- and low-risk patients. It is shown
that the patients in the high-risk group had a higher probability of death than those in the
low-risk group (Fig. 4D). In addition, Kaplan–Meier survival analysis also confirmed that
the survival time of the high-risk group was significantly reduced compared to the low-risk
group (Fig. 4E, $P < 0.001$). The predictive performance of the model was evaluated using
a time-varying ROC curve, with the area under the curve (AUC) reaching 0.829 at 1 year,
0.770 at 2 years, and 0.760 at 3 years (Fig. 4F). Overall, these findings suggest that MRRS
was able to distinguish the prognosis of HCC patients in the TCGA database.

## Validation of MRRS in the ICGC cohort

To test the robustness of the model developed by the TCGA cohort, we validated the
model with HCC patients' data from the ICGC database. We calculated the risk score for
each patient in the ICGC cohort using the same formula. Based on the above median, we
divided HCC patients from the ICGC cohort into high-risk ($n = 214$) and low-risk ($n = 13$)
(Fig. 5A). In the ICGC cohort, the results of PCA and t-SNE analyses were similar to those
of TCGA (Figs. 5B–5C). The high-risk group had a worse prognosis than the low-risk

**Table 2 The genes that contribute to the prognosis of HCC patients.** Based on the optimal value of λ, 14 genes that contribute the most to the prognosis of HCC patients were identified.

| Gene | Coef |
|------|------|
| DLAT | 0.087527704 |
| SEPHS1 | 0.295255334 |
| ACADS | −0.111567502 |
| UCK2 | 0.197793632 |
| GOT2 | −0.014390167 |
| ADH4 | −0.029554667 |
| LDHA | 0.324459855 |
| ME1 | 0.052008356 |
| TXNRD1 | 0.010457298 |
| B4GALT2 | 0.043281636 |
| AK2 | 0.19745112 |
| PTDSS2 | 0.178350248 |
| CSAD | −0.023302247 |
| AMD1 | 0.02060711 |

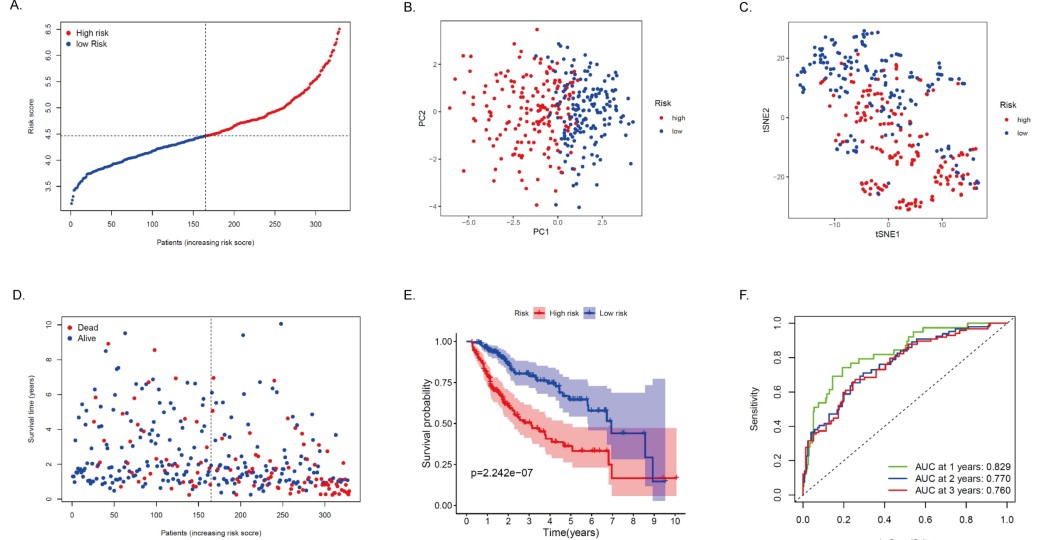

**Figure 4 Construction and verification of the MRRS in the TCGA cohort.** (A) The risk score distribution of HCC patients in the TCGA database. The color from blue to red indicates the level of expression from low to high. Scatter plots of high-risk and low-risk risk scores. (B) PCA analysis of CRA patients. (C) T-SNE analysis of CRA patients. (D) Survival status and survival time of patients. Red dots indicate patients who have died, and blue dots indicate that they are still alive. (E) Survival curves of low-risk and high-risk populations in HCC patients. Red represents the high-risk group, and blue represents the low-risk group. (F) 1-year, 2-year, and 3-year AUCs. AUC, the area under the curve; ROC, receiver operating characteristic.

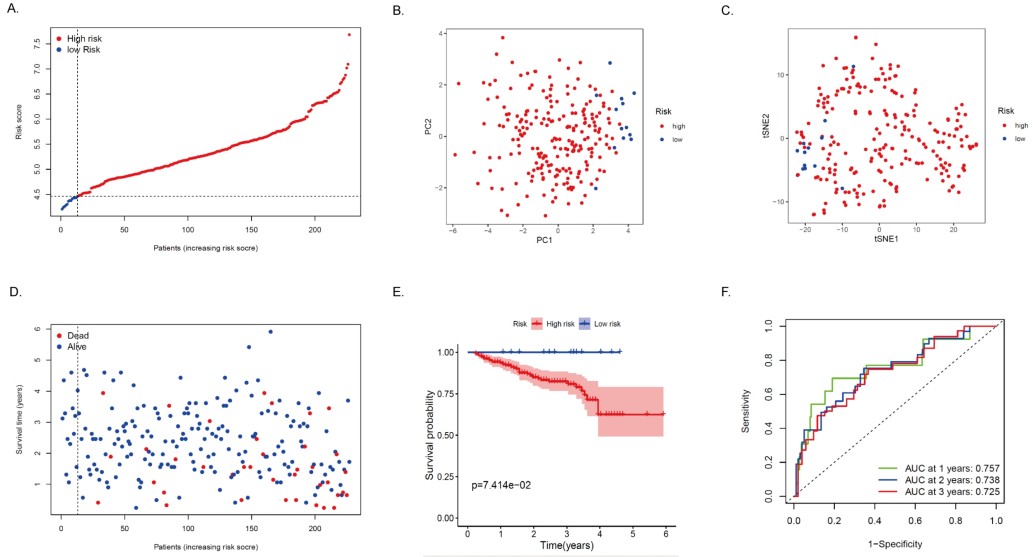

**Figure 5  Validation of the MRRS in the ICGC cohort.** (A) The risk score distribution of HCC patients in the ICGC database. The color from blue to red indicates the level of expression from low to high. Scatter plots of high-risk and low-risk risk scores. (B) PCA analysis of CRA patients. (C) T-SNE analysis of CRA patients. (D) Survival status and survival time of patients. Red dots indicate patients who have died, and blue dots indicate that they are still alive. (E) Survival curves of low-risk and high-risk populations in HCC patients. Red represents the high-risk group, and blue represents the low-risk group. (F) 1-year, 2-year, and 3-year AUCs. AUC, the area under the curve; ROC, receiver operating characteristic.

group (Fig. 5D). The survival curve showed that MRRS was effective in differentiating patients with different prognostic conditions, and patients in the high-risk group tended to have a higher risk of death and significantly shorter survival times than those in the low-risk group (Fig. 5E, $P < 0.05$). ROC analysis showed that MRRS had good predictive power on the prognosis of patients in the ICGC cohort (AUC = 0.757, 0.738, and 0.725; at 1, 2, and 3 years, respectively in ICGC, Fig. 5F). Taken together, this evidence shows that the model was able to predict the prognosis of HCC patients well.

## Independent prognostic value of MRRS in patients with HCC

To further determine whether MRRS can be used as an independent predictor of prognosis in HCC patients, we first performed univariate and multivariate Cox regression analysis in the TCGA cohort. As shown in Figs. 6A–6B, MRRS was found to be significantly correlated with the prognosis of HCC patients in both univariate Cox regression analyses (Hazard ratio (HR) = 3.757, 95% CI [2.819–5.007], $p < 0.001$) and multivariate Cox regression analysis (HR = 3.695, 95% CI [2.695–5.066], $p < 0.001$). Univariate and multivariate Cox regression analyses were then performed in the ICGC cohort for validation. The results showed that MRRS remained an independent predictor of prognosis in the ICGC cohort (HR = 3.023, 95% CI [1.958–4.667], $p < 0.001$, and HR = 2.574, 95% CI [1.602–4.135], $p < 0.001$; Figs. 6C–6D). In addition, the multivariate analysis to remove confounding factors, age, sex, and stage could not be used as independent prognostic indicators for HCC patients, showing the superiority of MRRS in the prognosis assessment of HCC patients.

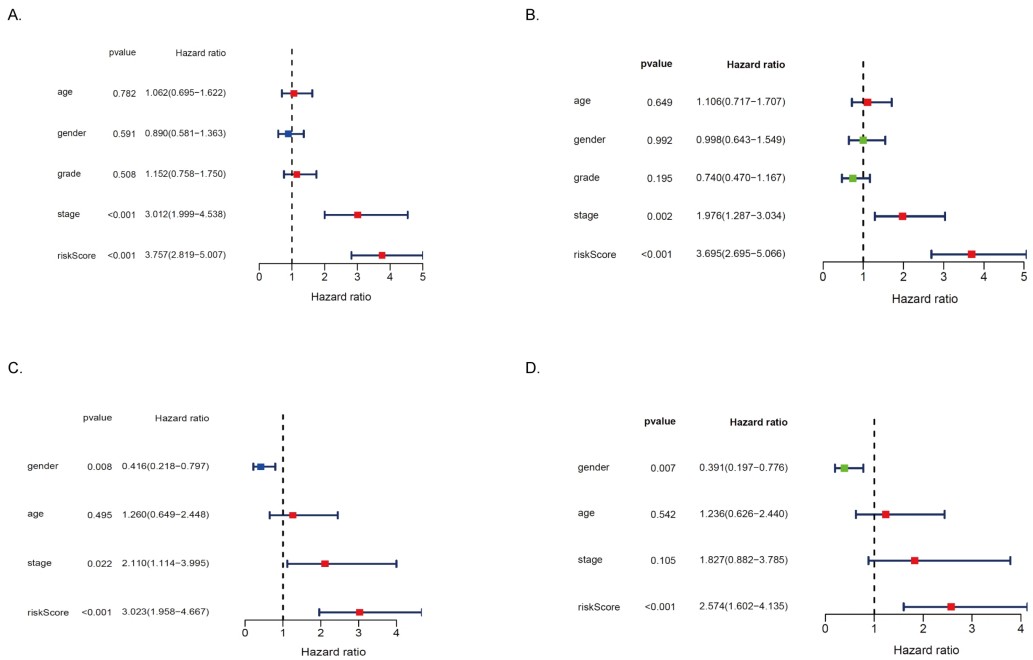

**Figure 6  MRRS is an independent prognostic signature for HCC patients.** (A) In the TCGA cohort, risk factors analysis of OS in the univariate Cox regression. (B) In the TCGA cohort, risk factors analysis of OS in the multivariate Cox regression. (C) In the ICGC cohort, risk factors analysis of OS in the univariate Cox regression. (D) In the ICGC cohort, risk factors analysis of OS in the multivariate Cox regression.HR hazard ratio.

Overall, our study suggests that MRRS can serve as an independent prognostic factor for patients with HCC.

## GO and KEGG analysis of DEGs in high and low-risk groups in the TCGA and ICGC cohorts

To elucidate the biological functions and pathways associated with MRRS risk scores, we first used GO enrichment analysis and KEGG pathway analysis for DEGs between high and low-risk groups in the TCGA cohort. The results of GO enrichment analysis showed that in terms of Biological Processes, signal transduction, cell division, and apoptotic process were significantly enriched. In terms of cellular components, cytosol, cytoplasm, and plasma membrane were significantly enriched; In terms of Molecular Function, protein binding, identical protein binding, and ATP binding were significantly enriched (Fig. 7A). KEGG enrichment results showed significant enrichment of metabolic pathways such as drug metabolism-cytochrome P450, retinol metabolism, central carbon metabolism, and tyrosine metabolism. Cell cycle, glycolysis/gluconeogenesis, ECM-receptor interaction, and signaling pathways such as the HIF-1 signaling pathway and PPAR signaling pathway were also enriched (Fig. 7B). In addition, immune-related signaling pathways such as the IL-17 signaling pathway and the TNF signaling pathway were significantly affected (Fig. 7B). Then we performed GO enrichment analysis and KEGG enrichment analysis for DEGs between high and low-risk groups in the ICGC cohort. GO enrichment analysis showed significant

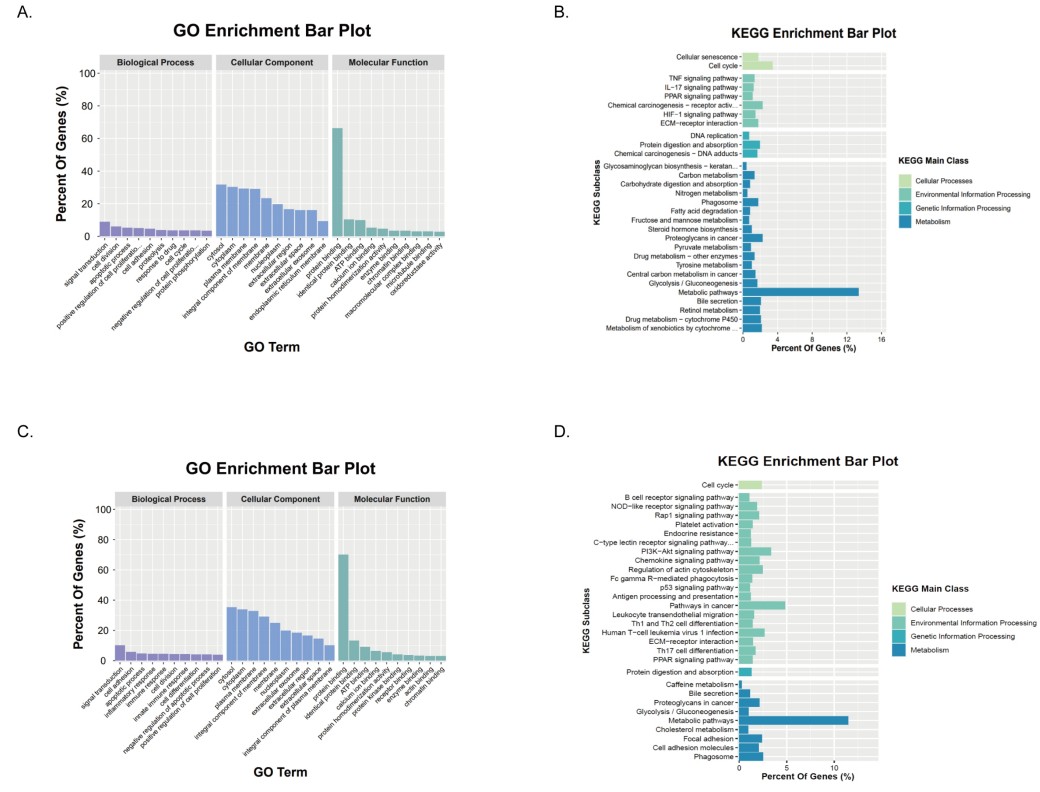

**Figure 7  Functional enrichment of the DEGs between risk groups.** (A–B) In the TCGA cohort, results of GO and KEGG pathway analysis. (C–D) In the ICGC cohort, results of GO and KEGG pathway analysis.

enrichment of cell adhesion, cell division, and positive regulation of T cell proliferation and activation pathways (Fig. 7C). KEGG enrichment results showed that in addition to the significant enrichment of metabolic signaling pathways, a variety of immune-related signaling pathways were also significantly enriched, such as Th17 cell differentiation, human T-cell leukemia virus 1 infection, Th1 and Th2 cell differentiation, leukocyte transendothelial migration, antigen processing and presentation, Fc gamma R-mediated phagocytosis, and B cell receptor signaling pathway (Fig. 7D). In addition, consistent with the results of the analysis in the TCGA cohort, the PPAR signaling pathway and PI3K-Akt signaling pathway were also significantly affected (Fig. 7D). Taken together, these findings highlight that HCC patients in the high and low-risk groups may lead differences in tumor malignant characterization through these pathways, ultimately leading to different prognoses of HCC patients.

## Infiltration of immune cells in TME in high- and low-risk populations

GO and KEGG enrichment analysis showed differences in function and pathways between risk groups in ECM-receptor interaction pathways and multiple immune-related signaling pathways. To further explore the correlation between different risk scores and immune status, we further explored the content of immune matrix components in the tumor

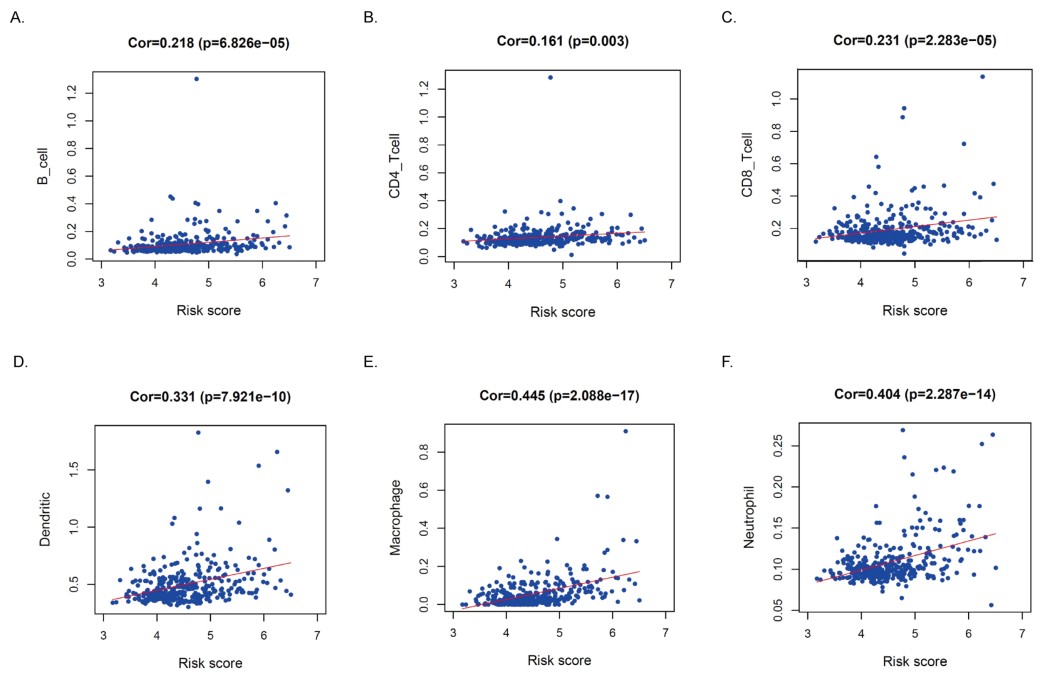

**Figure 8   Relationships between the MRRS and infiltration abundances of immune cells.** (A–F) The relationship between risk score and six types of immune cells.

microenvironment. According to the ESTIMATE algorithm, B cells, CD4+ T cells, CD8+ T cells, DC cells, macrophages, and neutrophil subsets in tumor tissues of high-risk groups were significantly upregulated significantly higher in the high-risk group (*P* < 0.05, Fig. 8), indicating that there are more immune cell components in TME in high-risk HCC patients.

## Downregulation of GOT2 promotes the migration capacity of hepatocellular carcinoma

By analyzing the relationship between prognosis and GOT2 mRNA expression levels in patients with hepatocellular carcinoma in the TCGA database, we found that patients with low GOT2 expression had a worse prognosis in the TCGA database (Fig. 9A). To further explore the biological significance of GOT2 in the progression of hepatocellular carcinoma, we first transfected siRNAs targeting GOT2 (siGOT2) or negative control siRNA (siNC) in HEK293 cells and verified their knockdown efficiency by RT-qPCR (Fig. 9B). The wound healing assay was used to assess the effect of GOT2 on the migration of hepatocellular carcinoma cells (Huh7 and MHCC97H). As shown in Figs. 9C–9F, down-regulation of GOT2 significantly inhibited migration in both cell lines compared to the control group. Together, our findings provide new insights into the role of GOT2 in influencing malignant phenotypes by regulating migration in hepatocellular carcinoma cells.

## DISCUSSION

HCC is an extremely common malignancy worldwide, it is important to develop reliable prognostic indicators for HCC patients. In this study, we established 14 genetic biomarkers

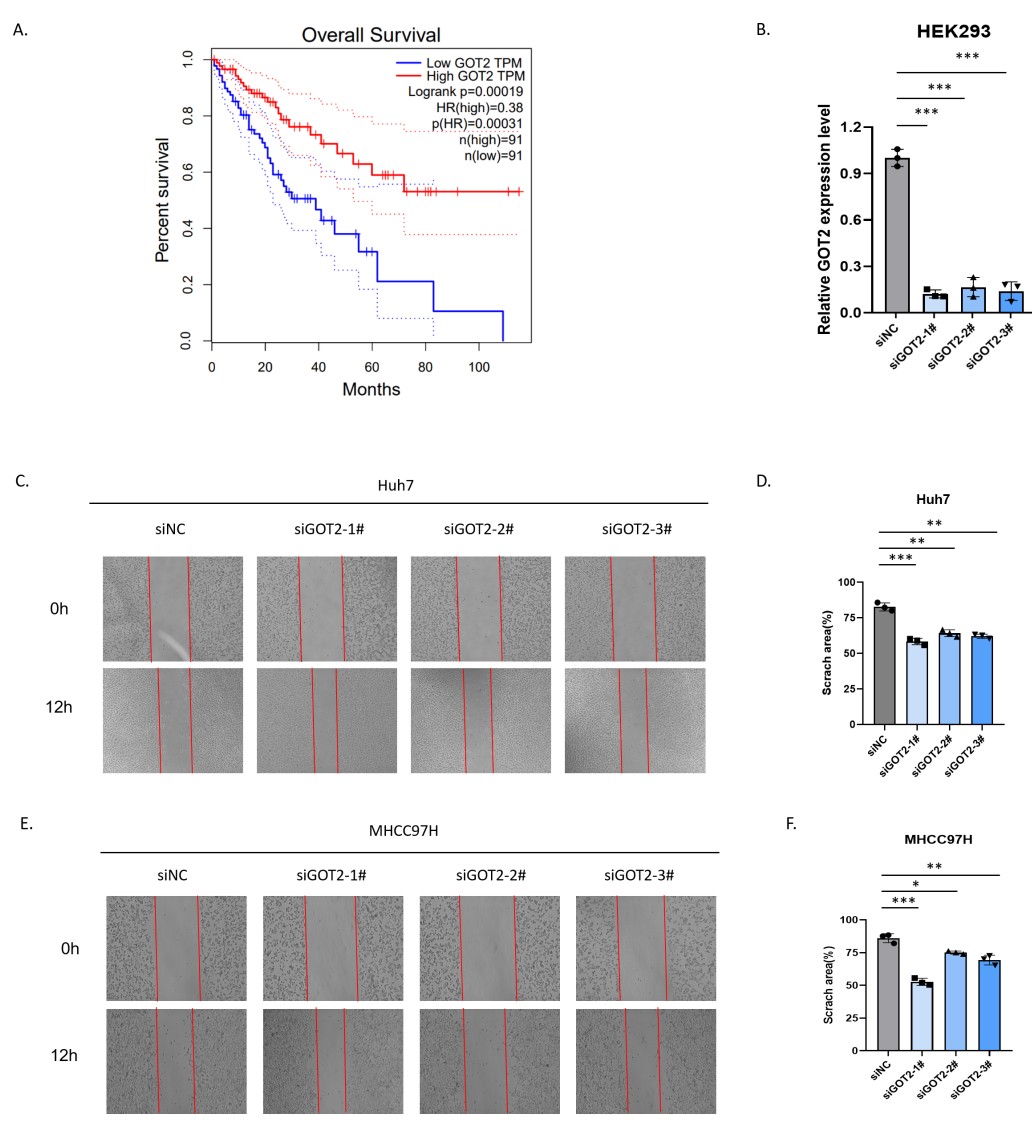

**Figure 9 Decreased expression of GOT2 in hepatocellular carcinoma inhibits migration.** (A) The Kaplan–Meier plots plot shows the relationship between GOT2 expression levels and patient outcomes. (B) HEK293 cell line was transfected with siNC or siGOT2 and the mRNA expression level of GOT2 was by RT-qPCR. (C–F) Wound healing assays showed that GOT2 reduced cell migration in Huh7 (C–D) and MHCC97H (E–F). Error bar indicates SD of the mean. $*p < 0.05$, $**p < 0.01$, $***p < 0.001$ by biological repeated-measures analysis of variance ($n = 3$).

as new prognostic models of MRRS and analyzed their ability to predict prognosis in HCC patients. The prognostic performance of the model is verified by the survival curve and ROC curve, and the results show that the model has good predictive performance. Its prediction efficiency was verified in the ICGC cohort. MRRS is a good independent indicator for predicting the prognosis of HCC patients in both TCGA and ICGC cohorts. Our prognostic model can help predict the prognosis of HCC patients clinically, thereby recommending better treatment measures for high-risk HCC patients.

GO and KEGG analysis showed that cell proliferation signals were significantly altered in patients in the high-risk group, suggesting that metabolism significantly affects cell proliferation, and cell proliferation may be the cause of poor prognosis in HCC patients. The mechanisms involved may be related to the PPAR signaling pathway, and PI3K-Akt signaling pathway, so PPAR inhibitors and PI3K-Akt inhibitors can be tried in high-risk patients to have a better prognosis for HCC patients. Functional analysis showed that immune signals were widely involved in tumor processes in high-risk patients, and more immune cells were infiltrated in the tumor microenvironment of high-risk patients. Our results are consistent with previous studies, and that increased expression of tumor blasts such as regulatory T cells, tumor-associated macrophages, tumor-associated neutrophils, and myeloid-derived suppressor cells in most cancers generally predicts worse outcomes (*Senovilla et al., 2012*). TME is generally divided into three categories (*Chen & Mellman, 2017*; *Lanitis et al., 2017*): (1) immune-inflamed: immune cells exist near tumor cells, (2) immune-excluded: immune cells exist around the stroma but do not penetrate the tumor, (3) immune-desert: lack of immune cell infiltration. In the current study, we reasonably speculate that tumors in patients with high-risk HCC may be Immune-excluded tumors (IETs). In this case, although TME shows abundant immune cell infiltration, cytotoxic T lymphocytes (CTL) cannot effectively infiltrate the tumor and exert a killing effect. This suggests that immunotherapy alone in high-risk patients may be less effective in patients. At present, the possibility that high-risk patients are immune-inflamed cannot be ruled out, and further evaluation of immunotherapy responses between different patients is needed to obtain better treatment outcomes. High-risk patients with immune inflammation should be aggressively treated with immunotherapy to achieve better clinical outcomes. This strategy may also be used in other cancer patients to seek better treatment targets and carry out precise treatment of cancer.

Although the mechanism of metabolism in tumors has become an increasingly popular area of research in recent years, the potential regulation between tumor immunity and metabolism still needs more research. Different states of tumor-associated macrophages (TAMs) are able to adapt to the tumor microenvironment by altering metabolism. It can inhibit the differentiation of M2-type TAMs by inhibiting the metabolism pathway (*Van den Bossche et al., 2016*; *Divakaruni et al., 2018*). Cancer-related MDSCs, whose main energy supply mode is converted from glycolysis to FAO. Increased fatty acid uptake and higher expression of key enzymes, which in turn upregulated the FAO rate necessary to produce immunosuppressive ARG1 (*Hossain et al., 2015*). Cytokines that drive the expansion of MDSCs. Fatty acid transporters (FATPs), as long-chain fatty acid transporters, upregulate and control the inhibitory activity of MDSCs on tumors in tumor-derived MDSCs (*Veglia et al., 2019*). This evidence suggests that fatty acid metabolism targeting MDSCs and M2 TAMs may be an important means of enhancing the efficacy of cancer immunotherapy. In this study, antigen presentation signaling pathways differed significantly between different risk groups. One possible hypothesis is that differences in mediators in tumor tissues attract antigen-presenting cells (APCs) to the site of tumor cells (*Friedmann Angeli, Krysko & Conrad, 2019*). Our study shows that many immune-related biological processes and pathways are enriched among HCC patients with different metabolism risk groups.

Therefore, it is reasonable to assume that metabolism in tumor tissues in HCC patients is closely related to tumor immunity.

Recent studies have found that amino acid metabolism (*Bertero et al., 2019*; *Ericksen et al., 2019*) and tricarboxylic acid metabolism (*Nie et al., 2020*) play an important role in tumor development and cancer drug resistance (*Yoo & Han, 2022*; *Ma et al., 2018*). As a member of glutamate-oxaloacetate aminotransferase, a recent study showed that GOT2 expression levels affect metabolism (*Wang et al., 2018*), suggesting that GOT2 may be an important regulator of cellular metabolism. In our study, the prognostic curve confirms that GOT2 plays an important role in predicting survival in patients with hepatocellular carcinoma. To gain a deeper understanding of the mechanism by which GOT2 affects tumor progression, wound healing assays are performed. These findings suggest that GOT2 has a significant activating effect on the migration ability of hepatocellular carcinoma.

In summary, our study defines a new prognostic model MRRS constructed from 14 MRGs. The model was shown to be independently correlated with OS in the derivation and validation cohorts, providing insights for predicting HCC prognosis. The study had several limitations. First, our prognostic model was constructed and validated with retrospective data from a public database. More forward-looking, real-world data are needed to validate its clinical utility. Secondly, since the model focuses on the evaluation ability of metabolism in HCC patients, a variety of indicators should be included in the comprehensive judgment of clinical decision-making. For example, it should be emphasized that the responsiveness of HCC patients in the high-risk score group to immunotherapy needs more research to elucidate.

## ACKNOWLEDGEMENTS

The authors would like to thank the TCGA, ICGC, and TIMER databases for the availability of the data.

### Funding

The authors received no funding for this work.

### Competing Interests

The authors declare that the research was conducted in the absence of any commercial or financial relationships that could be construed as a potential conflict of interest.

### Author Contributions

- Bin Ru conceived and designed the experiments, performed the experiments, analyzed the data, prepared figures and/or tables, and approved the final draft.
- Jiaqi Hu conceived and designed the experiments, performed the experiments, analyzed the data, prepared figures and/or tables, and approved the final draft.
- Nannan Zhang conceived and designed the experiments, analyzed the data, authored or reviewed drafts of the article, and approved the final draft.

- Quan Wan conceived and designed the experiments, authored or reviewed drafts of the article, and approved the final draft.

## Data Availability

The raw measurements are available in the Supplementary Files.

## Supplemental Information

Supplemental information for this article can be found online at http://dx.doi.org/10.7717/peerj.16335#supplemental-information.

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
