# Peer review of "A novel metabolism-related gene signature in patients with hepatocellular carcinoma"

_PeerJ, doi:10.7717/peerj.16335_

## Round 0.1 · original submission · Major Revisions

The most important issue is the contribution of the study. If the authors can address this issue, the manuscript can be further considered for publication. All the other issues have to be considered properly.

My decision will also depend on how the revision will be performed considering the grammar errors evidenced.

**Language Note:** The Academic Editor has identified that the English language must be improved. PeerJ can provide language editing services - please contact us at [email protected] for pricing (be sure to provide your manuscript number and title). Alternatively, you should make your own arrangements to improve the language quality and provide details in your response letter. – PeerJ Staff

Reviewer 1 ·

Basic reporting

Please see Additional Comments.

Experimental design

Please see Additional Comments.

Validity of the findings

Please see Additional Comments.

Additional comments

The authors conducted BOTH a series of bioinformatics analyses to work out a gene signature that could be useful for predicting prognosis of patients with hepatocellular carcinoma AND in vitro experiments to explore the role of GOT2 in hepatocellular carcinoma. The current study would be much improved if the authors address the following concerns:


------[Major Concerns about FIGURES, METHODS, RESULTS, and/or CONCLUSIONS]
1. This study's hypothesis — Lipid metabolism-related gene signatures could be useful for predicting prognosis and explaining hepatocellular tumorigenesis — has been investigated by some published papers (PMID: 33506899, PMID: 36456877, PMID: 36936948, PMID: 36421714, PMID: 37559997, PMID: 33579192, PMID: 32813933, PMID: 37346073, etc.). Furthermore, the current study used only TCGA databases (the same as those publications, especially PMID: 32813933 and PMID: 33579192) and did not further include other sequencing data. In addition, the role of GOT2 in hepatocellular carcinoma has also been reported by PMID: 35895805. Thus, it would be necessary to pinpoint the research/knowledge gap, which those published papers did not fill but the current study is addressing. In other words, please summarize the novelty of this study, compared with those similar publications.

2. In all FIGURES, it would be clear and more readable to BOTH provide figures with high resolution AND expand on figure legends by explaining the meanings of colors, groups, lines, and abbreviations. These revisions would greatly help readers to understand the results and their implications easily and efficiently. For example,
2.1 In all FIGURES' bar graphs, it would be more informative to display individual data points; in other words, please replace bar graphs by EITHER scatter plots with bars OR scatter plots (a pattern like PMID: 34537192, PMID: 37046252, and PMID: 37452367). Bar graphs have been shown to be misleading, because they cannot reveal variation/dispersion within data; instead, scatter plots with bars could be acceptable and scatter plots would be preferable (as confirmed by PMID: 25901488 and PMID: 28974579).
2.2 In all FIGURES' legends, it would be more rigorous to mention BOTH the sample size (the number of data points OR how many samples/patients were included) AND whether the data points were technical or biological replicates.
2.3 In all FIGURES' legends, it would be more rigorous to mention how the authors reported the data error (variation/dispersion): standard deviation (SD), confidence intervals (CI), or standard error of the mean (SEM, which would be not preferable).

3. Throughout this manuscript, the authors did not seem to define what were "lipid metabolism-related genes" and why the DEGs of 424 HCC patients in TCGA" were considered as "lipid metabolism-related genes". Such DEGs could be related to HCC but not necessarily associated with lipid metabolism. If the authors could not provide convincing explanations, please replace the expression "lipid metabolism-related genes" throughout this manuscript (including TITLE).

4. In ABSTRACT:
4.1 The authors seemed to split ABSTRACT into two paragraphs. It could be a routine to use only one paragraph.
4.2 It seems better to rewrite "Our research reveals that GOT2 is a lipid metabolism-related gene that affects the survival of patients with hepatocellular carcinoma ...", because the current study did not seem to provide direct evidence (in vitro experiments about lipid metabolism and/or GOT2-related randomized controlled trials) that EITHER GOT2 is related to lipid metabolism OR GOT2 affects patient survival. Thus, it would be more rigorous to change the sentence into "Our research reveals that GOT2 is negatively related to the survival of patients with hepatocellular carcinoma and GOT2 may contribute to tumor progression by inhibiting the ability of tumor cells to migrate".
4.3 It would be more informative to rewrite "Functional analysis showed that cell cycle-related signaling pathways were largely enriched" by clarifying BOTH what genes was analyzed by what types of "functional analysis".

5. In RESULTS:
5.1 It would be clearer to end each paragraph in RESULTS with one sentence: "Together, these results suggest that ..." (a pattern like PMID: 37452367, PMID: 34715879, PMID: 34384362, PMID: 35965679, and PMID: 34537192), summarizing a paragraph AND highlighting the implications of all results in the paragraph.


------[Minor Concerns about writing]
1. Throughout the manuscript, it seems better to use Grammarly (https://www.grammarly.com/) to check & correct potential grammatical errors or typos. For example,
1.1 In ABSTRACT ("Here, we aim to establish a novel lipid metabolism-related prognostic signature for predicting patients9 outcomes. To explore the value of LMRG expression in the prognosis prediction of hepatocellular carcinoma"), it seems better to change this sentence into "Here, we aim to ... outcomes and to explore the value of LMRG expression ...".
1.2 In ABSTRACT ("In our research, LMRRS was constructed consisting of 14 LMRGs"), it seems better to change this sentence into "In our research, a LMRRS model was constructed using 14 LMRGs".
1.3 In ABSTRACT, it would be clearer (easier to understand) to rewrite "According to the LMRRS model, the area under the curve (AUC) values predicted the prognosis of 1year, 3-year and 5-year prognosis of liver cancer patients reached 0.829, 0.760 and 0.739, respectively", which seems to have grammatical errors.

2. In ABSTRACT:
2.1 It would be more concise to delete the sentence "In recent years, it has already been a huge threat to human health worldwide", because it seems a repetition of the first sentence.
2.2 In "Low GOT2 expression is associated with poor prognosis in patients with hepatocellular carcinoma. Knocking down GOT2 significantly increased the migration capacity of hepatocellular carcinoma lines Huh7 and MHCC97H", it would be clearer and more cohesive (that is, sentences are closely connected) to rewrite these sentences and put them after "macrophages (p < 0.001) had a higher proportion of infiltrates in high-risk populations", after which the authors were intended to mention results about GOT2.
2.3 It would be more concise to delete the expression "and the migration capacity of siGOT2-1#, siGOT2-2# and siGOT2-3# increased by 235.5%, 66.7% and 95.7% compared with the control group".
2.4 It would be more informative and clearer (easier to understand) to mention the full name of "LMRRS" when this abbreviation comes first.

Reviewer 2 ·

Basic reporting

In the screening for MRGs that are associated with CRC prognosis, initially 75 IRGs with prognosis attributes were identified (Figure 3). Later, these number were reduced to 14. Why is there a difference here? Clarification is required to explained how 75 became 14.

The first two sentences of the first paragraph in discussion are redundant. It does not address the results of the study and is merely an introduction. The discussion section is too lengthy suggest streamlining the statements.

Experimental design

The authors defined the differential expressed genes as “Differential expression analysis was conducted, with an adjusted false discovery rate (FDR) < 0.05 as the thresholds.”. Which method did they use to calculate the p-value then FDR? Did they consider the batch effects in TCGA RNA-seq data?

Where are the references (literature support) to substantiate the statement that studies have reported on the relationship between lipid metabolism genes and HCC carcinogenesis in the introduction of the manuscript? There are several recent articles on predicting and identifying diagnoses by single cell and genomic sequencing. It is the continuous research that promotes better prediction and characterization of hepatocellular carcinoma and thus the clinical benefit of liver cancer patients.

Validity of the findings

Could the authors use some visualization method to show the P-value along with the Forest plot?

---

## Round 0.2 · accepted · Accept

The authors have performed the revisions as suggested by the Reviewers. The manuscript is now ready for publication.

Reviewer 1 ·

Basic reporting

Thank the authors for responding to all of my comments. The current version has been much improved.

Experimental design

N/A

Validity of the findings

N/A

Additional comments

N/A

Reviewer 2 ·

Basic reporting

well revision

Experimental design

well revision

Validity of the findings

well revision